# The mechanism of NDM-1-catalyzed carbapenem hydrolysis is distinct from that of penicillin or cephalosporin hydrolysis

Han Feng[1], Xuehui Liu [1], Sheng Wang[2], Joy Fleming[3,4], Da-Cheng Wang[1] & Wei Liu[5]

New Delhi metallo-β-lactamases (NDMs), the recent additions to metallo-β-lactamases (MBLs), pose a serious public health threat due to its highly efficient hydrolysis of β-lactam antibiotics and rapid worldwide dissemination. The MBL-hydrolyzing mechanism for carbapenems is less studied than that of penicillins and cephalosporins. Here, we report crystal structures of NDM-1 in complex with hydrolyzed imipenem and meropenem, at resolutions of 1.80–2.32 Å, together with NMR spectra monitoring meropenem hydrolysis. Three enzyme-intermediate/product derivatives, $EI_1$, $EI_2$, and EP, are trapped in these crystals. Our structural data reveal double-bond tautomerization from $\Delta^2$ to $\Delta^1$, absence of a bridging water molecule and an exclusive β-diastereomeric product, all suggesting that the hydrolytic intermediates are protonated by a bulky water molecule incoming from the β-face. These results strongly suggest a distinct mechanism of NDM-1-catalyzed carbapenem hydrolysis from that of penicillin or cephalosporin hydrolysis, which may provide a novel rationale for design of mechanism-based inhibitors.

[1] National Laboratory of Biomacromolecules, CAS Center for Excellence in Biomacromolecules, Institute of Biophysics, Chinese Academy of Sciences, Beijing 100101, China. [2] Key Laboratory of Molecular Biophysics of the Ministry of Education, College of Life Science and Technology, Huazhong University of Science and Technology, Wuhan, Hubei 430074, China. [3] Key Laboratory of RNA Biology, Institute of Biophysics, Chinese Academy of Sciences, Beijing 100101, China. [4] School of Stomatology and Medicine, Foshan University, Foshan, Guangdong 528000, China. [5] Institute of Immunology, The Third Military Medical University, Chongqing 400038, China. Correspondence and requests for materials should be addressed to D.-C.W. (email: dcwang@ibp.ac.cn) or to W.L. (email: wei.liu.2005@gmail.com)

β-lactam antibiotics have long been the most broadly used chemotherapeutic agents against bacterial infections[1]. All these compounds contain a four-membered β-lactam ring in their structural core and act as substrate analogs but actual long-lived inhibitors of bacterial transpeptidases, essentially blocking the cross-linking of adjacent peptidoglycan chains during cell wall biosynthesis[2]. However, the immoderate use of β-lactams during past decades has led to the evolution and spread of β-lactamases, a large enzyme family that efficiently catalyzes hydrolysis of the amide bond in the β-lactam ring and irreversibly inactivates antibiotics, including penicillins, cephalosporins, and carbapenems (Fig. 1)[1,3].

Based on amino acid sequence homology, β-lactamases are categorized into four classes, A, B, C, and D[4]. Classes A, C, and D are referred as serine-β-lactamases (SBLs), as they utilize a Ser residue to hydrolyze the β-lactam ring via an enzyme-acyl intermediate. Enzymes in class B are metallo-β-lactamases (MBLs) with Zn(II) ions present at the active site, which mediate hydrolysis without proceeding via a covalent intermediate[5,6]. MBLs are further divided into three subclasses, all adopting a similar αβ/βα fold and a common metal-binding motif through sharing low-sequence homology[7–10]. Although mechanism-based inhibitors of SBLs, such as clavulanic acid, tazobactam, and sulbactam, have been used clinically in combination therapies with β-lactam antibiotics[2], mechanism-based inhibitors against MBLs are currently unavailable in clinical settings due to poor understanding of the mechanisms underlying the hydrolysis of various substrates[10].

New Delhi metallo-β-lactamases (NDMs) are the most recent additions to the class of MBLs[11,12]. The emergence of this novel plasmid-encoded MBL family heralds a new era of antibiotic resistance due to their ability to hydrolyze almost all clinically available β-lactam antibiotics and rapid worldwide dissemination. Their highly efficient inactivation of the last-generation carbapenems, such as imipenem and meropenem[13], is of particular concern, as carbapenems are regarded as "antibiotics of the last resort" due to their resistance to many SBLs and broader spectrum of activity than other lactams. Since identification of the first NDM-type lactamase, NDM-1, in 2009, kinetic[14], spectroscopic[15,16], crystallographic[17–20], and computational[21,22] studies

and combined investigations using multiple techniques[23,24] have attempted to uncover the hydrolytic mechanism. NDM-1 belongs to the B1 subclass of MBLs that requires a dinuclear metal center for full catalytic activity. Zn1 is ligated to three histidine residues, H120, H122, and H189, while Zn2 is coordinated with D124, C208, and H250.

As the mechanism is currently understood, MBL-mediated hydrolysis is believed to proceed via two steps: cleavage of the amide bond and protonation of the generated intermediate[8,10,25,26]. After the formation of a Michaelis complex (ES), a water/hydroxide molecule residing between the two Zn(II) ions acts as a nucleophile to attack the carbonyl carbon (C7) and cleave the C–N bond. In synchrony with the opening of the β-lactam ring, an anionic intermediate is generated with the newly formed carboxylate binding to Zn1 and the amide nitrogen (N4) and the carboxylate of the β-lactam-fused ring interacting with Zn2 (EI). In the following step, the intermediate is protonated, and an EP complex is tentatively formed before product release from the enzyme pocket. A significant body of experimental evidence indicates that decay of the anionic intermediate in EI is the rate-limiting step in a turnover of the antibiotics[7,8,10].

Although the overall reaction steps are known, it is unclear if all hydrolysable bicyclic β-lactams, including those with distinct chemical structures, are hydrolyzed by the same general mechanism. In penicillin hydrolysis, the intermediate is thought to contain a negative charge on the lactam nitrogen (N4)[8,10]. This charge in an anionic intermediate of cabarpenems or cephalosporins, however, is delocalized over a conjugated π-system encompassing the double bond in the lactam-fused pyrroline or dihydrothiazine ring, possibly resulting in double-bond rearrangement from position 2–3 to 3–4 (3–4 to 4–5 in cephalosporins) (Fig. 1). The resultant carbanionic intermediate has been detected in the hydrolysis of imipenem, nitrocefin, and chromacef catalyzed by NDM-1 and other B1 MBLs by spectroscopic studies[16,27,28]. We recently reported more solid evidence based on crystal structures of NDM-1 in complex with hydrolyzed cefuroxime and cephalexin. Both structures revealed sp3 hybridization of C3 in the cephalosporoate intermediates, clearly indicating double-bond tautomerization and formation of a carbanionic intermediate in EI[24].

Similarly, carbapenems usually undergo pyrroline tautomerization from $\Delta^2$ to $\Delta^1$, i.e., a double-bond shift from position 2–3 to 3–4 when hydrolyzed by SBLs, as revealed in published spectroscopic data and crystal structures[29–31]. Carbapenem hydrolysis by MBLs, however, is relatively poorly studied. The only report implying pyrroline tautomerization is the detection of two diastereomeric products in BcII-catalyzed imipenem hydrolysis[28]. To date, crystallographic evidence of tautomerization in hydrolyzed carbapenems catalyzed by MBLs is lacking.

Another intriguing question on the matter of much debate is the protonation source for β-lactam intermediates. The proton may be donated from a metal-bound water, bulky solvent, or even the newly formed carboxylic acid[18,21–23]. In all reported structures of NDM-1 in complex with hydrolyzed penicillins or cephalosporins, there is a water molecule bridging the two Zn(II) ions independent of Zn1–Zn2 distances, suggesting that a terminal water molecule originally bound to Zn2, or an incoming solvent molecule promptly takes up the vacant position originally occupied by the nucleophilic hydroxide after β-lactam ring opening[10]. This bridging water molecule is regarded as an ideal proton donor[8,10,26]. However, it does not exist in the available NDM-1 structures in complex with hydrolyzed meropenem (PDB entries 4EYL[19] and 4RBS).

To address whether carbapenems undergo pyrroline tautomerization during hydrolysis and whether diverse protonation mechanisms exist for different β-lactams, we conduct a combined

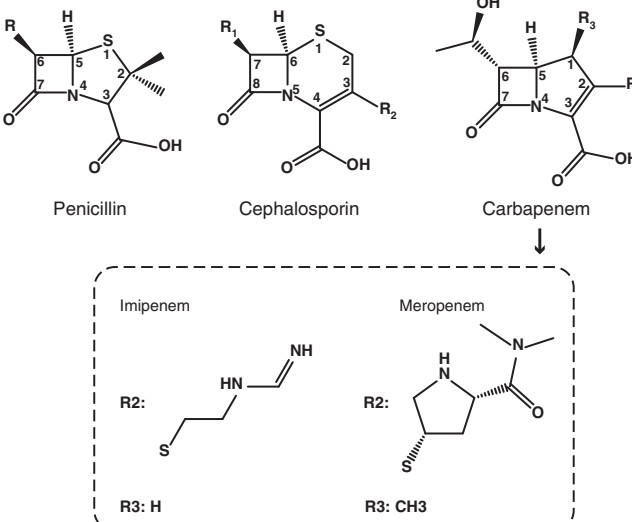

**Fig. 1** Chemical structures of penicillin, cephalosporin, and carbapenem. The chirality of C6 is *S* in carbapenems, contrary to *R* at the same position in penicillins and cephalosporins. The R2 and R3 side chains of imipenem and meropenem are shown in the dashed rectangle

**Table 1 Crystallographic data collection and refinement statistics of NDM-1 in complex with hydrolyzed imipenem**

|  | Crystal 1 | Crystal 2 | Crystal 3 | Crystal 4 | Crystal 5 | Crystal 6 |
|---|---|---|---|---|---|---|
| *Data collection* | | | | | | |
| Space group | $P2_1$ | $P2_1$ | $P2_12_12_1$ | $P2_12_12_1$ | $P2_1$ | $P2_12_12_1$ |
| Cell dimensions | | | | | | |
| $a, b, c$ (Å) | 69.24, 73.61, 154.52 | 69.38, 74.10, 154.52 | 69.35, 73.93, 77.49 | 69.77, 74.01, 77.43 | 69.90, 74.10, 155.63 | 69.22, 73.76, 77.43 |
| $\alpha, \beta, \gamma$ (°)[a] | 90, 90.083, 90 | 90, 90.259, 90 | 90, 90, 90 | 90, 90, 90 | 90, 90.321, 90 | 90, 90, 90 |
| Resolution (Å)[a] | 2.0 (2.05–2.0) | 2.3 (2.36–2.30) | 1.95 (2.0–1.95) | 2.0 (2.05–2.0) | 2.0 (2.05–2.0) | 1.8 (1.84–1.8) |
| $R_{sym}$[a] | 0.072 (0.440) | 0.080 (0.457) | 0.052 (0.244) | 0.085 (0.592) | 0.111 (0.593) | 0.070 (0.555) |
| $I/\sigma(I)$[a] | 8.65 (2.06) | 8.32 (2.06) | 15.06 (3.17) | 11.21 (2.70) | 7.10 (2.14) | 16.80 (2.55) |
| Completeness (%)[a] | 90.0 (61.6) | 95.2 (97.3) | 94.1 (70.4) | 99.8 (99.8) | 96.4 (98.8) | 94.0 (64.8) |
| Redundancy[a] | 1.73 (1.37) | 1.94 (2.00) | 3.45 (2.19) | 3.80 (3.82) | 2.43 (2.43) | 6.36 (4.44) |
| *Refinement* | | | | | | |
| Resolution (Å) | 47.96–2.0 | 48.17–2.3 | 42.36–1.95 | 42.46–2.0 | 48.48–2.0 | 42.28–1.8 |
| Reflections | 100,028 | 69,446 | 28,575 | 27,718 | 105,994 | 35,229 |
| $R_{work}/R_{free}$ | 0.1858/0.2285 | 0.1779/0.2372 | 0.1695/0.2172 | 0.1994/0.2344 | 0.2025/0.2342 | 0.1685/0.2038 |
| No. atoms | | | | | | |
| Protein | 13,600 | 13,592 | 3398 | 3387 | 13,600 | 3406 |
| Ligand | 168 | 168 | 42 | 42 | 216 | 42 |
| Water/Ion | 1954 | 1860 | 317 | 339 | 1871 | 377 |
| *B*-factors | | | | | | |
| Protein | 25.44 | 37.56 | 29.23 | 41.07 | 27.29 | 28.07 |
| Ligand | 35.44 | 49.65 | 46.11 | 47.7 | 34.81 | 45.46 |
| Water/Ion | 35.51 | 41.56 | 37.97 | 42.98 | 37.54 | 38.63 |
| R.m.s. deviations | | | | | | |
| Bond lengths (Å) | 0.005 | 0.004 | 0.004 | 0.005 | 0.006 | 0.008 |
| Bond angles (°) | 0.905 | 0.849 | 0.877 | 0.965 | 1.013 | 1.134 |
| Ramachandran plot | | | | | | |
| Favored (%) | 98.4 | 98.67 | 98.89 | 96.23 | 97.73 | 99.12 |
| Allowed (%) | 1.6 | 1.33 | 1.11 | 3.1 | 2.1 | 0.88 |
| Disallowed (%) | 0 | 0 | 0 | 0.67 | 0.17 | 0 |
| Captured complex | $EI_2$ ($\Delta^1$) | $EI_1$ ($\Delta^2$) | EP ($\Delta^1$) | $EI_2$ ($\Delta^1$) | $EI_2$ ($\Delta^1$) | EP ($\Delta^1$) |
| PDB ID | 5YPK | 5YPI | | | | 5YPL |

[a]Values in parentheses are for highest-resolution shell

investigation on NDM-1-catalyzed hydrolysis of imipenem and meropenem (Supplementary Fig. 1) using X-ray crystallography and NMR spectrometry. The experimental data obtained in this study suggest that the mechanism of carbapenem hydrolysis is very likely different from the mechanism established based on experimental data obtained from penicillin and cephalosporin hydrolysis, particularly in the step of intermediate decay, and possibly provide a novel rationale for designing mechanism-based inhibitors against MBLs.

## Results

**Pyrroline tautomerization observed in crystal structures**. NDM-1 was crystallized in space group $P2_12_12_1$ using a previously reported condition[24], and ca. 30 complex crystals were prepared by soaking with excess imipenem or meropenem for different durations, followed by flash cooling and X-ray data collection. Six imipenem-bound and five meropenem-bound structures were determined by molecular replacement and refined at resolutions from 1.80 to 2.32 Å (Tables 1, 2). Roughly half of these crystals were indexed in space group $P2_1$ (eight monomers in the asymmetric unit) rather than $P2_12_12_1$ (two monomers) with the β angle slightly deviating from 90°, probably reflecting marginal lattice cracks in some crystals after soaking. Nonetheless, clear omit $F_o – F_c$ electron density for potential ligands above 3.0 σ was observed at the active site in each crystal, suggesting good incorporation of the carbapenem substrates.

We first modeled hydrolyzed imipenem or meropenem in the $\Delta^2$ isoform (Supplementary Fig. 1) in all crystals and found that the ligands could be well fitted in density in some structures with good planarity remaining among C1, C2, C3 and the sulfur atom bonded to the R2 side chain during refinement (Fig. 2a, d; Supplementary Figs. 3, 6). In the other structures, however, imperfect ligand fitting was observed as the blob of density surrounding C2 in the refined models puckered upward with respect to its bonded atoms, suggesting $sp^3$ hybridization of this atom. We then tried to model the $\Delta^1$ tautomer (Supplementary Fig. 1) in those structures, which resulted in much better density fitting after 10 cycles of full-model refinement (Fig. 2b, c, e; Supplementary Figs. 4, 5, 7), strongly indicating the correctness of such modeling. We finally modeled $\Delta^2$ ($EI_1$) in one imipenem- and three meropenem-bound structures and $\Delta^1$ ($EI_2$ or EP) in five imipenem- and two meropenem-bound structures (Tables 1, 2). While both isomers have previously been observed in a few SBL–carbapenem complex structures[30,32–35], here they are crystallographically trapped for the first time in reactions catalyzed by an MBL.

The carbapenem molecules showed occupancy from 0.69 to 1.0 after refinement, and all displayed a well-resolved structural core including the pyrroline ring, C2-sulfur, and the hydroxyethyl moiety. The sulfur-bonded pyrrolidine ring in meropenem was also well defined in density (Fig. 2d, e), but the hydrophilic formimidamide group in imipenem was rather mobile, as indicated by higher *B*-factors or even the absence of visible density (Fig. 2a–c). The difference in flexibility of the R2 side chains in imipenem and meropenem revealed here is consistent with the counterparts present in the reported SBL carbapenem structures[34,35].

**Conserved conformation between tautomers**. Conformational changes upon pyrroline tautomerization, including reorientation of the β-lactam carbonyl and/or the hydroxyethyl side chain, were

**Table 2 Crystallographic data collection and refinement statistics of NDM-1 in complex with hydrolyzed meropenem**

|  | Crystal 7 | Crystal 8 | Crystal 9 | Crystal 10 | Crystal 11 |
|---|---|---|---|---|---|
| *Data collection* | | | | | |
| Space group | $P2_12_12_1$ | $P2_1$ | $P2_12_12_1$ | $P2_12_12_1$ | $P2_1$ |
| Cell dimensions | | | | | |
| $\quad a, b, c$ (Å) | 68.73, 73.51, 76.53 | 69.79, 74.02, 155.04 | 69.79, 73.72, 77.50 | 69.79, 74.01, 77.27 | 70.16, 74.12, 155.43 |
| $\quad \alpha, \beta, \gamma$ (°) | 90, 90, 90 | 90, 90.437, 90 | 90, 90, 90 | 90, 90, 90 | 90, 90.321, 90 |
| Resolution (Å)[a] | 2.12 (2.18–2.12) | 2.15 (2.21–2.15) | 2.32 (2.38–2.32) | 1.95 (2.00 – 1.95) | 2.32 (2.38–2.32) |
| $R_{sym}$[a] | 0.109 (0.621) | 0.101 (0.399) | 0.092 (0.627) | 0.100 (0.651) | 0.094 (0.513) |
| $I/\sigma(I)$[a] | 10.66 (2.18) | 7.1 (2.11) | 9.85 (2.06) | 8.93 (2.12) | 11.23 (2.13) |
| Completeness (%)[a] | 99.8 (99.8) | 95.4 (93.9) | 99.6 (99.8) | 99.5 (60.8) | 98.9 (89.5) |
| Redundancy[a] | 3.85 (3.52) | 2.12 (1.89) | 3.45 (3.51) | 3.74 (3.71) | 3.44 (2.83) |
| *Refinement* | | | | | |
| Resolution (Å) | 41.98–2.12 | 48.34–2.15 | 42.42–2.32 | 42.46–1.95 | 48.48–2.32 |
| Reflections | 22,608 | 85,215 | 20,273 | 29,801 | 68,817 |
| $R_{work}/R_{free}$ | 0.2039/0.2439 | 0.1959/0.2254 | 0.2161/0.2751 | 0.2027/0.2283 | 0.1668/0.2045 |
| No. atoms | | | | | |
| $\quad$ Protein | 3398 | 13,596 | 3402 | 3379 | 13,592 |
| $\quad$ Ligand | 54 | 216 | 54 | 54 | 216 |
| $\quad$ Water/Ion | 176 | 1389 | 201 | 444 | 1163 |
| *B*-factors | | | | | |
| $\quad$ Protein | 35.39 | 29.34 | 54.99 | 35.60 | 28.59 |
| $\quad$ Ligand | 56.75 | 44.63 | 76.00 | 55.72 | 38.68 |
| $\quad$ Water/Ion | 37.03 | 36.40 | 47.17 | 43.76 | 34.70 |
| R.m.s. deviations | | | | | |
| $\quad$ Bond lengths (Å) | 0.003 | 0.004 | 0.004 | 0.004 | 0.004 |
| $\quad$ Bond angles (°) | 0.995 | 0.874 | 0.947 | 1.074 | 0.946 |
| Ramachandran plot | | | | | |
| $\quad$ Favored (%) | 98.45 | 98.67 | 97.57 | 98.0 | 98.51 |
| $\quad$ Allowed (%) | 1.55 | 1.33 | 2.43 | 1.56 | 1.22 |
| $\quad$ Disallowed (%) | 0 | 0 | 0 | 0.44 | 0.28 |
| Captured complex | $EI_2$ ($\Delta^1$) | $EI_1$ ($\Delta^2$) | $EI_2$ ($\Delta^1$) | $EI_1$ ($\Delta^2$) | $EI_1$ ($\Delta^2$) |
| PDB ID | 5YPN | 5YPM | | | |

[a]Values in parentheses are for highest-resolution shell

observed in the crystallized enzyme-acyl adducts of SBLs, such as TEM-1, AmpC, SHV-1, and BlaC, with imipenem, meropenem, or ertapenem bound at the active site[32,36–38]. Unlike those structures, the carbapenem molecules revealed in our structures showed good conformational convergence between the two iso-forms (Fig. 2f) and even between the two substrates (Fig. 2g). Except for the flexible R2 side chain, the structural core in all imipenem or meropenem copies could be well overlaid. The newly formed C6-carboxylate interacts with Zn1 in a consensus conformation stabilized by a hydrogen bond from the side chain of N220 (Fig. 3a–c), while the hydroxyethyl groups are consistently oriented with the hydroxyl moiety approaching D124. Such highly conserved conformations among carbapenem copies clearly indicate that the double-bond rearrangement within the pyrroline ring induced little conformational change in hydrolyzed substrates. This feature seems to be a noteworthy difference between MBL-mediated hydrolysis and SBL-catalyzed reactions.

With regard to the enzyme, the amino acids essential for substrate binding, such as W93, K211, and N220, remain invariant positions and orientations among our and the previously reported NDM-1 structures[18,19,24] (Supplementary Fig. 2a, b). The only exception is the β-hairpin loop L3, which is rather mobile in the available structures, as reflected by labile loop conformations and alternative side chain rotamers of the amino acids therein, such as F70. Compared with the penicillin- or cephalosporin-bound structures, this loop moves closer to the active site in our structures and the PDB entry 4EYL[19], resulting in a narrower substrate-binding groove (Supplementary Fig. 2c). This difference is probably attributable to the smaller size of the R1 side chain in carbapenems (Fig. 1).

**Crystallographic capture of three reaction complexes.** Although both tautomers showed convergent conformations, our structures revealed two divergent modes of imipenem-zinc binding. In four imipenem- and all five meropenem-bound structures determined in this study, the carbapenem intermediates present in either $\Delta^2$ ($EI_1$) or $\Delta^1$ ($EI_2$) bind the dinuclear metal center in a consensus manner with hydrolyzed penicillins or cephalosporins revealed in NDM-1 structures[18,19,23,24]. The newly formed C6-carboxylate contacts Zn1, while the lactam nitrogen (N4) and C3-carboxylate interact with Zn2 (Fig. 3a, b; Supplementary Fig. 8a, b). However, in the remaining two structures containing $\Delta^1$ imipenem, one oxygen atom of the C6-carboxylate (C7O) intercalates between the two Zn(II) ions, giving rise to hexahedral coordination of Zn2 with a significantly decreased C7O–Zn2 distance, from 3.2 to 2.5 Å (averaged for all imipenem copies) (Fig. 3c; Supplementary Figs. 5, 8c). The imipenem molecules in these two structures show an evident shift forward to Zn2 by ~0.5 Å with respect to their counterparts in $EI_2$, albeit remaining an almost identical overall conformation (Fig. 3d). As a result of this translational shift, the C3-carboxylate moves farther away from Zn2, markedly increasing the C3O–Zn2 distance to 3.3 Å, essentially beyond a canonical ligand-metal interacting range. Although N4 still maintains tight contact with Zn2 in this binding mode, the enzyme-intermediate interaction must be weakened due to the detachment of the C3-carboxylate from Zn2.

In comparison with other imipenem-bound structures, the antibiotic molecules modeled in these two structures have significantly lower overall occupancy than the imipenem counter-parts in other structures (0.74 vs. 0.89) and display higher flexibility in the R2 side chain (Fig. 3c). Considering the

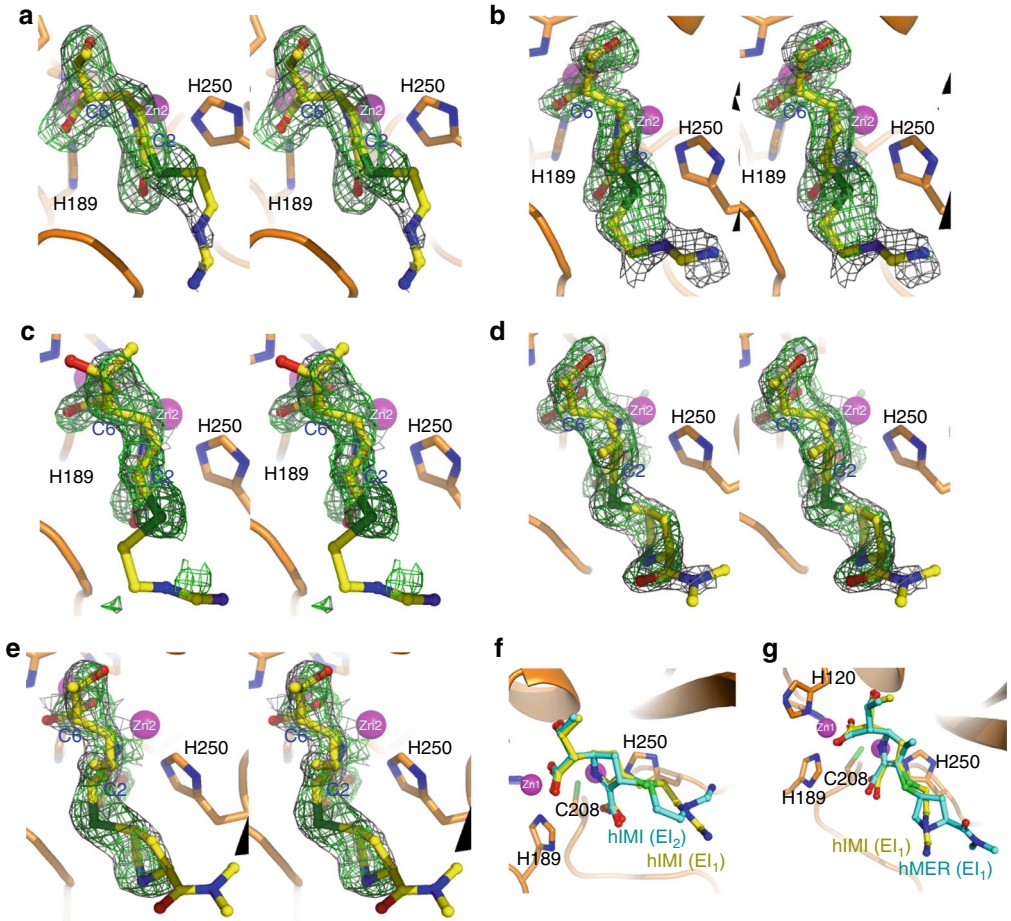

**Fig. 2** Imipenem and meropenem intermediates bound to the active site of NDM-1. **a–e** Stereo views of the $\Delta^2$ isomer of hydrolyzed imipenem (**a**), or meropenem (**d**), in EI$_1$, the $\Delta^1$ isomer of hydrolyzed imipenem (**b**), or meropenem (**e**), in EI$_2$, and the $\Delta^1$ isomer of hydrolyzed imipenem in EP (**c**), represented as ball-and-stick models. The $2F_o - F_c$ electron density shown as gray mesh is contoured at 1.0 σ and the omit $F_o - F_c$ electron density shown as green mesh is contoured at 3.0 σ. **f** Overlay of hydrolyzed imipenem in EI$_1$ and EI$_2$. **g** Overlay of hydrolyzed imipenem and meropenem in EI$_1$. Zinc ions are represented by magenta spheres in all panels

weakened imipenem–metal interaction and lower refined occupancy, we suppose that the species revealed in these two structures more likely represent an EP derivative in which a protonated product has been generated and is ready to be released from the active site.

In this study, we failed to capture such a species in meropenem-bound structures, but structural superimposition of imipenem in EP and meropenem in the published structure (PDB entry 4EYL)[19] showed good overlay, particularly the C6-carboxylate with one oxygen atom intercalating between the Zn (II) ions (Fig. 3e). The similar mode of zinc binding as well as the low-ligand occupancy suggests that the NDM-1–mereopenem complex present in that structure may represent a similar EP derivative as that shown in Fig. 3c. This surmise is further supported by superimposition with meropenem in EI$_2$ determined in this study, which showed a similar translational shift of the hydrolyzed antibiotics forward to Zn2 in 4EYL (Fig. 3f).

**Absence of the bridging water molecule.** All NDM-1 structures with penicillin or cephalosporin substrates bound at the active site have revealed a highly ordered water molecule residing between the Zn(II) ions that is proposed to be an ideal proton donor for anionic intermediates[10,21,22]. However, there is no such water molecule in the available structures containing hydrolyzed

meropenem (4EYL and 4RBS)[19]. Consistent with those structures, electron density corresponding to this water molecule was not observed in our structures (Fig. 3a–c). This means that the lack of the bridging water molecule is thus very likely a common feature in carbapenem-bound structures and is apparently independent of composition or protonation states of the intermediate within the enzyme pocket.

A structural determinant for the presence/absence of the bridging water molecule was unraveled by mutual comparisons of available NDM-1 structures in complex with different hydrolyzed β-lactam substrates. As a characteristic feature of carbapenems, the chirality of C6 is *S*, contrary to *R* at the same position in penicillins and cephalosporins (Fig. 1). The resultant *trans* configuration of the C5–C6 bond in carbapenems forces the newly formed C6-carboxylate to orient differently from its counterpart in hydrolyzed cephalexin (Fig. 3g), cefuroxime (Fig. 3h), and ampicillin (Supplementary Fig. 8d, e). As a consequence of such an orientation, the two oxygen atoms of this carboxylate moiety in imipenem or meropenem are positioned in bidentate interacting distance with Zn1 (Fig. 3a–c). Notably, one atom is inevitably located within a distance shorter than 1 Å from the position presumably occupied by the bridging water (Fig. 3g, h; Supplementary Fig. 8d, e). Such an orientation of the C6-carboxylate resulting from the unique stereochemical feature of carbapenems would obviously lead to steric hindrance with a

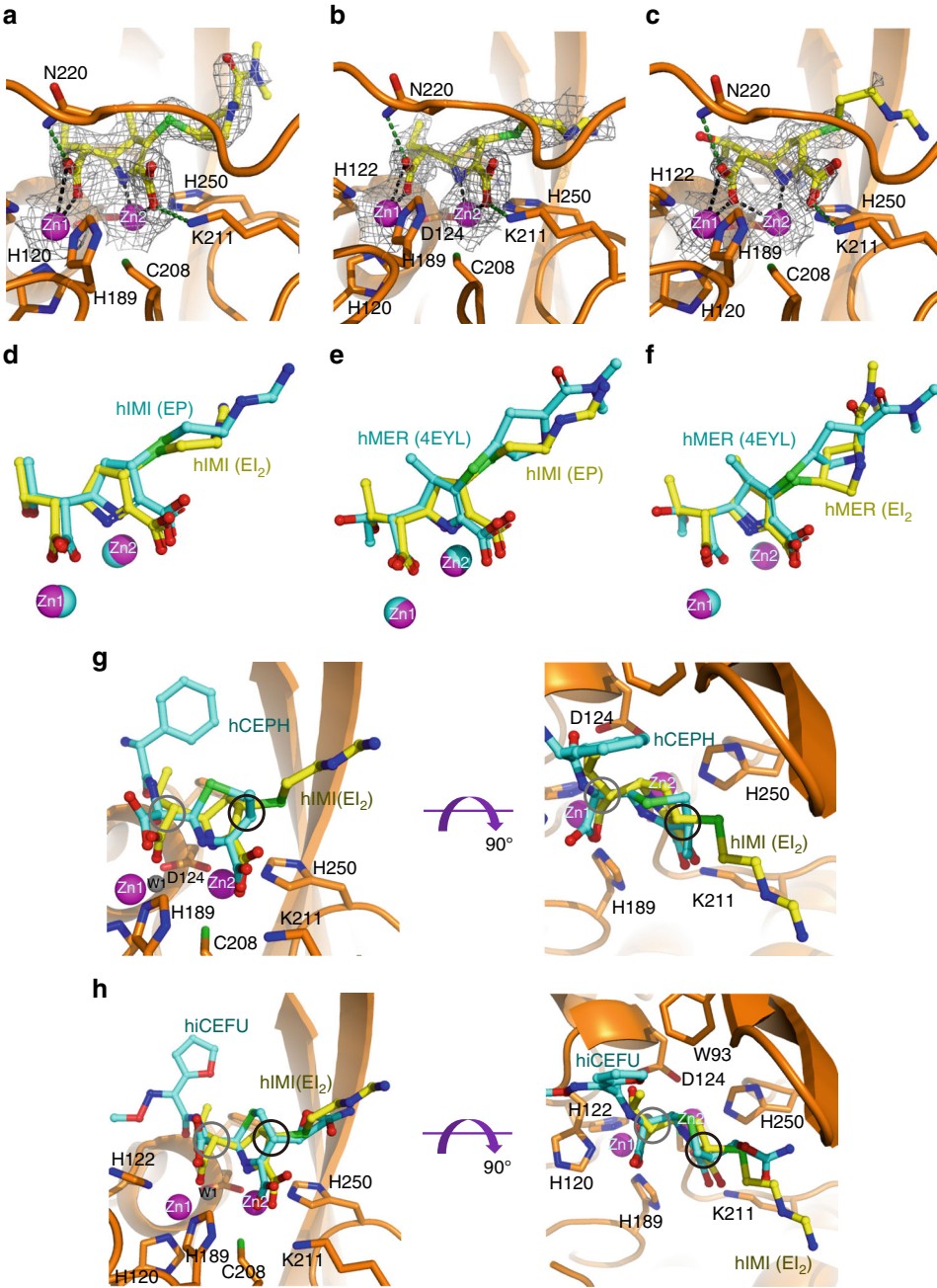

**Fig. 3** Close-up views of the active site of NDM-1 representing different enzyme-intermediate/product adducts. **a–c** Zinc centers bound by hydrolyzed meropenem in EI1 (**a**), or imipenem in EI2 (**b**), or EP (**c**). The $2F_o - F_c$ electron density is contoured at 1.0 σ and shown as gray mesh. Metal coordination bonds and hydrogen bonds with N220 and K211 are denoted by dashed lines in black and green, respectively. **d–f** Overlay of imipenem in EI2 and EP (**d**), imipenem in EP, and meropenem in structure 4EYL[19] (**e**), or meropenem in EI2 and that in structure 4EYL[19] (**f**). **g, h** Overlay of hydrolyzed imipenem in EI2 with cephalexin (**g**), or cefuroxime (**h**)[24]. C2 and C6 in imipenem are highlighted by black and gray circles, respectively. Zinc ions are represented by magenta spheres in all panels

water molecule potentially sitting between the Zn(II) ions, accounting for the absence of the bridging water in carbapenem-bound structures.

**Single epimer of hydrolyzed carbapenems in the Δ¹ isoform.**
Following the pyrroline tautomerization that was clearly observed in our structures, a new chiral center is formed in the carbanionic intermediate upon sp³ hybridization of C2 (Fig. 2). This raises an intriguing question regarding the stereoselectivity of intermediate

protonation. Two possible epimers are expectedly generated if protonation occurs in aqueous milieu, where there is almost an equal chance for proton uptake from either face of the antibiotic molecule. However, if the intermediate is protonated when it is still bound at the active site, only a single epimer with proton input from a given face is likely generated due to steric hindrance. In BcII-mediated hydrolysis of imipenem, for example, two diastereomeric products were detected in a 7:3 ratio, indicating the existence of a stereoselective advantage for one epimer over the other[28].

By contrast, all imipenem or meropenem molecules in $\Delta^1$ (EI$_2$ or EP) present in our structures displayed $S$ chirality at C2 (Supplementary Fig. 1). This overwhelming selective advantage is rather surprising to us since different configurational isomers have been captured in the crystal structures of NDM-1 with cephalosporins[24]. The generated epimer is consistent with hydrolyzed cefuroxime (Fig. 3h) but contrary to hydrolyzed cephalexin with proton uptake solely from the α face[39] (Fig. 3g). Since the cefuroxime intermediate snapshot in the crystal structure was a species prior to the carbamoyl group leaving and contained a negative charge at C3, the $S$ chirality of this atom might represent a preferred stereoselection with a lone pair of electrons. However, carbapenems that lack such a leaving group must be protonated at C2 after double-bond tautomerization.

Given that it is impossible to assess the protonation state of $\Delta^1$ intermediates by X-ray crystallography, two possibilities may exist: (I), they remain unprotonated, and the current epimer represents an intrinsically preferred chiral selection upon sp³ hybridization of C2; or (II), they are protonated in the crystal, implying proton uptake from the β face only. The crystallographic capture of imipenem in EP gives us good confidence in assuming the protonated state of this species due to its pre-released conformation and lower occupancy. However, we tend to presume that the intermediates in EI$_2$ are unprotonated, since they displayed significantly higher occupancy, and an anionic intermediate is expected to bind Zn(II) more tightly than a protonated species. The single epimer displaying $S$ chirality at C2 observed in hydrolyzed imipenem and meropenem lead us to speculate that, at least in crystallo, the carbapenem intermediates are protonated prior to their release from the metal center, and it is very likely a bulky solvent molecule from the β face donates the proton.

**Exclusive β-diastereomeric product detected by NMR.** To exclude possible crystallographic artifacts, we needed to confirm if the exclusive chiral selectivity observed in our crystal structures can also be detected in carbapenem hydrolysis proceeded in solution. To this end, we monitored the NDM-1 catalyzed hydrolysis of meropenem by $^1H$ and $^{13}C$ NMR using a previously established protocol[24]. The $^1H$ spectrum was characterized by time-dependent development of a single proton signal ($\delta$ 3.87, s) within 20 min after the start of hydrolysis (Fig. 4a; Supplementary Table 2), indicating input of a proton during the reaction. As expected, the chemical shift, intensity, and multiplicity corresponding to this signal allowed us to assign it to a proton bonded to C2, which was further confirmed by a series of 2D spectra (Methods). Consistently, the $^{13}C$ spectra showed a significant downfield shift of C2 from $\delta$ 140.67 to 59.74 and an upfield shift of C3 from $\delta$ 136.09 to 176.17 induced by the hydrolysis (Supplementary Fig. 9). All these spectroscopic data undoubtedly indicate a double-bond shift from position 2–3 to 3–4.

To verify the stereoselectivity of hydrolyzed meropenem, a $^1H$-$^1H$ ROESY spectrum was collected on completion of the reaction. A strong cross signal arising between H2 and the pyrroline methyl hydrogen (H10) and a much weaker signal between H2 and H1 were present in the spectrum (Fig. 4b), suggesting that H2 is closer to H10 than to H1 in space. The NMR data obtained in this study positively indicate that a β-diastereomer is generated during meropenem hydrolysis. This is further verified by the absence of a cross peak between H2 and H5, a hydrogen atom located at the α face (Fig. 4b). Notably, α-diastereomeric product was not detectable in our NMR experiments, agreeing well with the exclusive epimer showing $S$ chirality at C2 revealed in our crystal structures. The good convergence of crystallographic and spectroscopic data provides solid evidence supporting in situ protonation of the $\Delta^1$ intermediate at the active site rather than in

aqueous milieu, which disallows the generation of an α-diastereomer probably associated with the missing bridging water molecule.

## Discussion

The mechanism of MBL-catalyzed carbapenem hydrolysis is inadequately understood relative to that of penicillin and cephalosporin hydrolysis, largely due to difficulties in characterizing the molecular structure of hydrolytic intermediates using X-ray crystallography, NMR, or other spectroscopies[9,10]. Our data presented here revealed informative details of the reaction including (i) double-bond tautomerization in the pyrroline ring following the cleavage of the lactam amide bond; (ii) three derivatives presumably representing EI$_1$, EI$_2$, and EP; (iii) absence of the bridging water molecule attributed to steric hindrance from the newly formed carboxylate that is forced to take a distinctive orientation resulting from the *trans* configuration of the β-lactam ring in carbapenems; and (iv) exclusive stereoselective protonation of the carbanionic intermediates.

A branched reaction mechanism was proposed by Tioni et al.[28] based on the detection of double diastereomeric products of BcII-hydrolyzed imipenem by Raman spectroscopy. The crystallographic and NMR spectroscopic data obtained in this study, however, allowed us to propose a linear reaction mechanism of NDM-1-mediated carbapenem hydrolysis (Fig. 5), as only a single diastereomeric product was detected (Fig. 4b). The crystallographic entrapment of both $\Delta^2$ and $\Delta^1$ isomers sufficiently demonstrates that hydrolysis proceeds via an open-ring derivative with a negative charge delocalized over a conjugated π-system covering C2 and N4, which is dynamically stabilized at the active site of NDM-1. A tentative equilibrium between the two intermediates, EI$_1$ and EI$_2$, is thus established before protonation occurs. Owing to the lack of the bridging water that may serve as an ideal proton donor, protonation at the lactam nitrogen (N4) seems to be impossible. This scenario obviously favors EI$_2$ accumulation if the $\Delta^2$ isomer cannot decay unless it is tautomerized to $\Delta^1$.

An imipenem species presumably in EP was fortuitously captured in two crystal structures. Both the lower occupancy and weakened interactions with the dinuclear metal center of NDM-1 suggest that the hydrolyzed antibiotic is protonated and ready to be released from the enzyme pocket (Fig. 3c) and further imply that protonation occurs while an intermediate in EI$_2$ is still bound to the active site. This hypothesis is convincingly supported by our NMR data showing the presence of only a single diastereomeric product generated from hydrolysis in solution. This experimental result undoubtedly rules out the possibility of intermediate protonation in aqueous milieu, which would otherwise give rise to two possible diastereomers.

There are two possible pathways from EI$_2$ to EP. A transient species might be generated if the intermediate is protonated in the canonical conformation as the species in EI$_2$ (EI$_3$ in Fig. 5), which is similar to hydrolyzed cephalexin bound at the active site of NDM-1[24]. Alternatively, the intermediate could undergo a translational shift with respect to the zinc ions and adopt the conformation observed in EP before being protonated, which gives rise to another transient species (EP' in Fig. 5). Unfortunately, it is difficult to decipher the more likely pathway because hydrogen atoms are usually invisible in X-ray structures, making it impossible to assess the protonation state of trapped intermediates. Neutron diffraction and quantum chemistry calculation may be suitable approaches for addressing this question.

Taken together, the crystallographic snapshots and NMR spectra obtained in this study uncover common features in carbapenem and cephalosporin hydrolysis such as double-bond

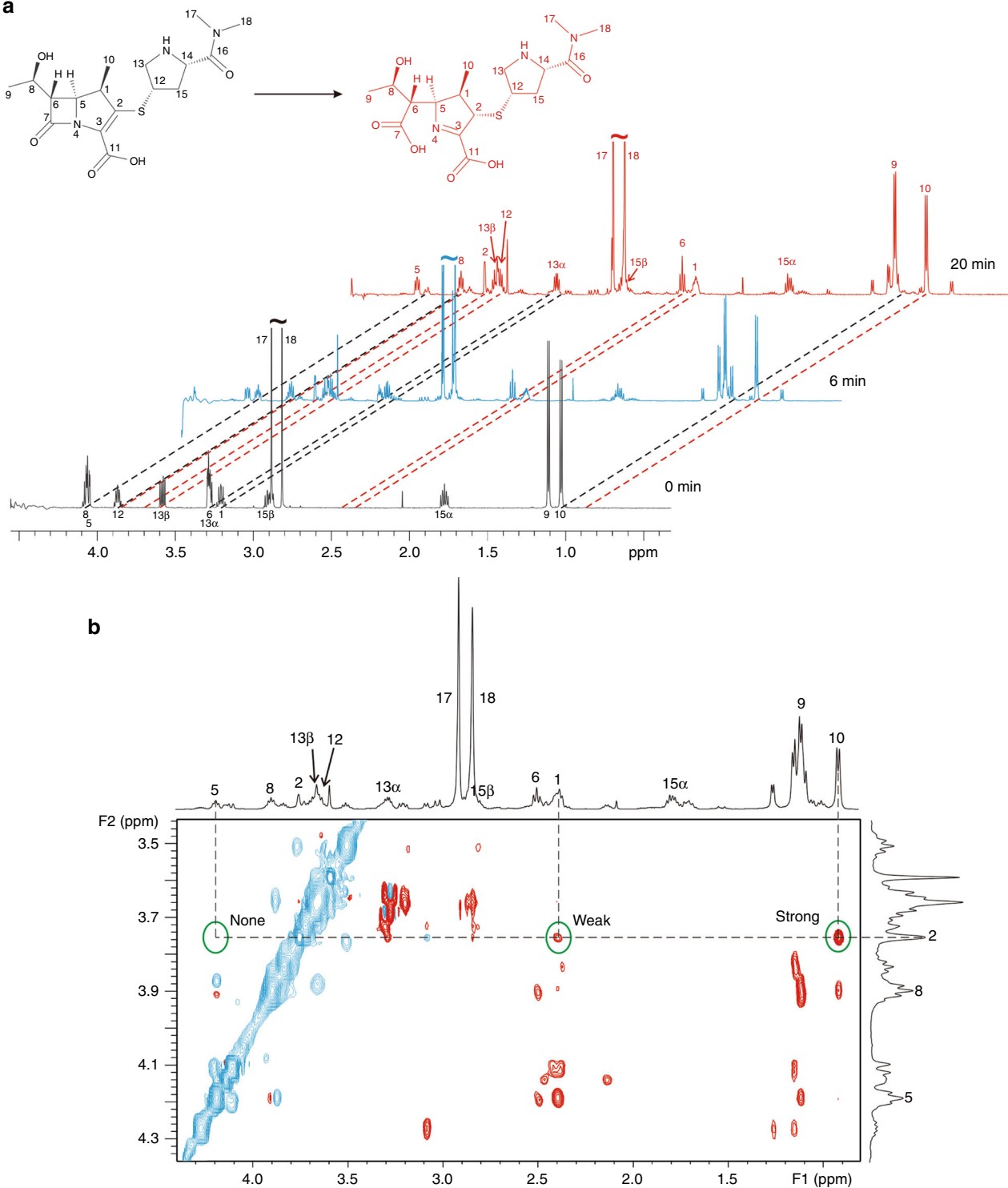

**Fig. 4** NMR spectra monitoring meropenem hydrolysis catalyzed by NDM-1. **a** $^1$H NMR spectrum of hydrolyzed meropenem recorded before and 6 or 20 min after NDM-1 addition to the reaction system. **b** Part of a ROESY spectrum of the hydrolysis product. Diagonal and cross peaks are shown in blue and red, respectively. Proton signal assignments are labeled beside the peaks. The chemical shifts of H2, H1, H5, and H10 are highlighted by dashed lines

tautomerization arising from π-conjugated negative charge delocalization within the β-lactam-fused pyrroline or dihydrothiazine ring. Our study also reveals unique properties suggesting that the mechanisms underlying carbapenem and cephalosporin hydrolysis have distinct features. One noteworthy difference is the lack of a bridging water molecule at the active site bound to the hydrolyzed carbapenems (Fig. 3g, h), which may significantly

affect the process of intermediate protonation. In fact, the stereoselective epimers generated in imipenem/meropenem hydrolysis different from that in cephalexin hydrolysis might result from such an effect. Proton uptake exclusively from the β face firmly suggests that the carbapenem intermediate is protonated by a bulky solvent molecule entering into the pocket from the exterior space. On the other hand, the absence of the bridging

**Fig. 5** Hypothesized reaction mechanism of carbapenem hydrolysis catalyzed by NDM-1. The $EI_1$, $EI_2$, and EP complexes are trapped in crystal structures and the β-diastereomeric product is detected by NMR. Two other possible intermediates, $EI_3$ and EP', are not experimentally characterized. The pink spheres represent zinc ions and the surrounding labels are zinc-coordinating amino acids

water molecule further precludes the possibility of proton donation from an apical water molecule bound to Zn2 or an incoming solvent molecule approaching the metal center. Both of these scenarios require Zn(II)-mediated deprotonation, probably through proton relay with the bridging water/hydroxide molecule, and would definitely give rise to an α-face protonation[21,23,40], similar to hydrolyzed cephalexin[24]. Conclusively, the mechanism of carbapenem and cephalosporin hydrolysis diverges in the rate-limiting step of intermediate protonation, as the proton comes likely from different donors. This seems to be closely correlated with whether the bridging water molecule exists or not, which in turn solely depends on the intrinsic molecular structures of antibiotic substrates.

Since the first characterization of NDM-1 in 2009, 16 variants displaying comparable carbapenemase activities have been identified in this family. These variants differ from NDM-1 by one or two amino-acid substitutions, none of which are involved in zinc coordination, substrate interaction, or maintenance of the active site conformation[11,12]. We therefore reason that the catalytic

mechanism proposed here can be extrapolated to other NDM-typed MBLs.

We noted that the ratio of $\Delta^2$ ($EI_1$) to $\Delta^1$ ($EI_2$ or EP) was 1:5 in imipenem-bound structures but 3:2 in meropenem-bound structures (Tables 1, 2), which probably means that there are inconsistent stereoselective advantages between the two carbapenem substrates. More $\Delta^1$ present in hydrolyzed imipenem may hint at a faster rate of pyrroline tautomerization, in agreement with the larger $k_{cat}$ value for imipenem than that of meropenem in NDM-1-mediated hydrolysis[13,14,41]. It is possible that the formimidamide group in imipenem is more favorable for intermediate protonation, as it provides a more hydrophilic milieu for a bulky solvent molecule to approach, while the relatively hydrophobic pyrrolidine ring in meropenem is somewhat unfavorable. If this is true, the chemical nature of the R2 side chains may serve as an important determinant of the turnover rate of carbapenem substrates. In this sense, side chain modification might be a strategic consideration for the rational design of mechanism-based inhibitors against MBLs.

## Methods

**Protein expression and purification.** A coding sequence for full-length NDM-1 (Supplementary Table 1) was synthesized using GENEWIZ (Sangon Biotech, Shanghai, China). The nucleotide sequence encoding an N terminus truncated protein (residues 29–270) was amplified with a pair of primers (Supplementary Table 1) by PCR. The amplified product was later inserted into a pET28a vector between *Nde*I and *Xho*I restriction sites. The recombinant protein with an N-terminal His$_6$-tag was produced in *Escherichia coli* strain BL21(DE3) (Merck, Germany) at 22 °C with an incubation for 16–20 h after induction with 0.5 mM isopropyl β-D-1-thiogalactopyranoside (IPTG). Harvested cells were resuspended in the lysis buffer containing 50 mM Tris pH 8.0, 500 mM NaCl, 10 mM imidazole, 10 mM β-mercaptoethanol, and 1 mM ZnCl$_2$. Bacteria were lysed by sonication on ice at 200 W using 3-s pulses with 7-s intervals for 16.5 min before the removal of insoluble debris by centrifugation for 30 min at 13,000 × g and 4 °C. The supernatant was immediately loaded onto a Ni$^{2+}$-NTA chromatography column (Novagen), followed by column washing and elution with 250 mM imidazole added in the same buffer. The His$_6$-tag was subsequently removed with thrombin digestion for 12 h at 4 °C prior to reloading the protein solution onto the same Ni$^{2+}$-NTA column. NDM-1 without the His$_6$-tag was eluted in the flow-through fraction. After the fractions were pooled, the protein was further purified by anion exchange chromatography using a HiTrap Q 5-ml column (GE healthcare) and size exclusion chromatography using a HiLoad superdex75 16/600 column (GE healthcare). The purified protein was stored in 20 mM Tris pH 8.0, 150 mM NaCl, and 2 mM DTT and frozen at −80 °C until further use.

**Crystallization and diffraction data collection.** The purified NDM-1 protein was concentrated to 30 mg ml$^{−1}$ before crystallization trails. All crystallization experiments were carried out using the hanging-drop vapor-diffusion method at 20 °C, with each drop formed by mixing 1 μl protein solution and 1 μl reservoir solution before equilibration against 500 μl reservoir solution. Crystals of NDM-1 were obtained under a previously reported condition containing 28% (w/v) PEG3350, 0.1 M Bis-Tris, pH 5.8, and 0.2 M ammonium sulfate[24]. Complex crystals with imipenem or meropenem were obtained by crystal soaking. The antibiotics were added into the drops containing NDM-1 crystals with a molar ratio of 1:5 (protein: antibiotic) before incubation at room temperature for 10 min − 8 h.

Immediately after soaking, the crystals were transferred into 100% paraffin oil for 10 s before flash cooling in the stream of liquid nitrogen. Diffraction data were collected on beamline BL18U1 and BL19U1 at Shanghai Synchrotron Radiation Facility (SSRF), China, with a wavelength of 0.97853 Å. Imipenem-bound crystals were determined at resolutions of 1.8–2.3 Å, and meropenem-bound crystals, at 1.95–2.32 Å. All X-ray data were indexed, integrated, and scaled using XDS[42].

**Structure determination and refinement.** All structures were determined by the molecular replacement using Phaser[43], with chain A of the cefuroxime-bound NDM-1 structure (PDB entry 4RL0)[24] as a search model. The structures were refined using *Phenix.refine*[44] with several rounds of manual remodeling in Coot[45] between refinement cycles. Models of hydrolytic imipenem and meropenem in either Δ$^2$ or Δ$^1$ tautomer (Supplementary Fig. 1) were built manually using the ligand builder embedded in Coot[46]. The resultant models were sent to the ProDRG2 server[47] to generate topological dictionary files, which were later edited manually before using them in subsequent refinement. After manual modeling of the ligands in an omit difference density map ($F_o − F_c$) contoured at 3.0 σ, several cycles of full-model refinement were conducted. The final model was validated using MolProbity[48]. Statistics of data collection and structure refinement are summarized in Tables 1 and 2. All figures showing structure representations were prepared using the molecular visualization program Pymol[49].

**NMR experiments for monitoring the hydrolysis of meropenem.** NMR samples of meropenem were prepared by dissolving the antibiotic compounds in aqueous solvent of H$_2$O/D$_2$O (90%/10% v/v). The catalytic reaction was initiated by mixing 50 nM NDM-1 and 500 μM meropenem in 20 mM phosphate buffer, pH 5.9, and 150 mM NaCl. The hydrolytic process was monitored by $^1$H-NMR spectrometry every 2 min after the start of reaction. To facilitate the signal-to-atom identification, $^{13}$C NMR spectra were collected before hydrolysis was initiated and after the reaction was completely finished. All NMR experiments were carried out at room temperature using an Agilent 600 MHz spectrometer. The assignment of the NMR signals was performed using both the $^1$H and $^{13}$C spectra and validated with a set of 2D experiments including $^1$H-$^1$H distortionless enhancement by polarization transfer spectrum (DEPT), $^1$H-$^1$H correlation spectrum (COSY), $^1$H-$^1$H total correlation spectrum (TOCSY), $^1$H-$^{13}$C heteronuclear single quantum correlation spectrum (HSQC), $^1$H-$^{13}$C heteronuclear multiple bond correlation spectrum (HMBC), and $^1$H-$^1$H rotating-frame nuclear Overhauser effect spectrum (ROESY, mixing time of 200 ms). The $^1$H and $^{13}$C spectra were collected with a spectrum width of 9615.4 and 37878.8 Hz, with 16,384 and 26,515 complex data points collected for each scan, respectively. The spectrum widths of the 2D spectra were as follows, 5733.9 Hz for the $^1$H dimension and 22630.8 (HSQC) or 28663.6 Hz (HMBC) for the $^{13}$C dimension. While 1024 complex points were collected for the $^1$H dimension, 150 (for HSQC), 200 (for TOCSY, ROESY, and HMBC), or 256 (for COSY) complex data points were collected for the indirectly detected dimensions.

All proton chemical shifts were referenced to external DSS (0.00 ppm) and the carbon chemical shifts were referenced indirectly. All spectra were displayed and analyzed by using MestReNove software (version 6.1.0). The postscript file was exported using the VnmrJ software package (version 3.2), which was also the operating system for the spectrometer. Figures displaying NMR results were prepared using Corel Draw (version 12.0.0.458).

**Data availability.** Coordinates and structure factors have been deposited in the Protein Data Bank under the accession codes 5YPI (EI$_1$ of NDM-1-imipenem), 5YPK (EI$_2$ of NDM-1-imipenem), 5YPL (EP of NDM-1-imipenem), 5YPM (EI$_1$ of NDM-1-meropenem), and 5YPN (EI$_2$ of NDM-1-meropenem). Other data are available from the corresponding authors upon reasonable request.

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

## Acknowledgements

This work was supported by the Strategic Priority Research Program of the Chinese Academy of Sciences (XDB08020200). We thank the staffs of the BL18U/BL19U1 beamline at the National Facility for Protein Science Shanghai (NFPS) of Shanghai Synchrotron Radiation Facility, for assistance during data collection.

## Author Contributions

H.F. performed protein purification and crystallization, and X-ray data collection; X.L. conducted NMR spectroscopic measurements; S.W. analyzed the data; D.-C.W. conceived the study; W.L. designed the experiments, determined and refined the crystal structures, and wrote the manuscript. J.F. proofread the manuscript.

## Additional information

**Competing interests:** The authors declare no competing financial interests.

