## [Peer Review File · Nature Communications]

NATURE COMMUNICATIONS

Title: The mechanism for carbapenem hydrolysis catalyzed by NDM-1 is distinctive from penicillin or cephalosporin hydrolysis

Firstly, I should say that I think this work is worthy of publication and that it does *indicate* a different mechanism (or at least different kinetics) for MBL catalyzed carbapenem hydrolysis versus that of penicillins or cephalosporins. The new carbapenem derived complex structures are of interest – even though they are similar to those reported by the authors and other groups. The key think here is the lack of observation of a bridging water in the carbapenem derived structures, but this has already been observed.

What I *thought* the paper was going to describe was time-resolved studies on the reaction of carbapenems in crystals in which the crystal structures and spectroscopic studies in crystals are combined. This is not the case. The NMR studies simply analyse the products, not intermediates. The crystallography involves soaking – hence it cannot be ruled out that the crystallographically observed complexes arise from hydrolysed products (re)binding. I'm also unconvinced that at the resolution described the quality of the maps is sufficient to unequivocally define ratios of data 1/2 isomers – only to predict which is likely to predominate.

So overall, I'm not convinced this is a paper for Nature Communications. One option would be for the authors to publish the results in a specialist crystallography journal (which I'm supportive of). Alternatively, they should carry out spectroscopic studies (UV/vis and, maybe, IR/Raman) on the *same* crystals on which they collect time-resolved diffraction data in order to validate the density they observe as intermediates (though I appreciate this is non-trivial work), i.e., carry out time resolved studies. They should also rule out the possibility of products rebinding to the enzyme. So overall, I think the results are interesting, but that they are over-interpreted.

Abstract/Discussion:

Will need revision in the light of the comments above.

Introduction

L38 add citation.

L62 NDM-1 is not the most recent MBL – NDMs may be the most recent additional family – check.

L67 some SBLs do catalyse carbapenem hydrolysis.

L68 'are resistant to many SBLs'.

L 76-77 use standard BBL numbering throughout

L82 'proceed in two stages'?

L86 how can the anion be generated whilst the β -lactam is intact?

L91 whether anion intermediate protonation is rate limiting should depend on the substrate?

L104 add citation.

L106 the crystallography does inform in the protonation state at N1 or is this inferred?

L110-112 there are other reports, see e.g. Chem Sci 2015, 6, 956-963.

L134 note the tautomerization has been previously addressed – see ref 10.

L143-153 Giving soaking times / enzyme /lactam ratios.

L189-192 Evidence for the EP derivative needs to be supplied.

L287-288 Isn't it possible the structures are derived from products (re)binding to the enzyme, in which case one of the stereoisomers may bind more tightly, biasing the results.

L335-340 the key novel aspect is the stereo selective protonation – but this needs to be validated further, at least in terms of the crystallographic data.

Show mFo-Dfc map in figure 2 SI (not, just, 2Fo-fc map).

L143 I do think soaking with hydrolysed carbapenems should be tried.

L144 complex.

L176 how do the structures compare with the reported structure – in ref 18.

Comment on the hydroxyethyl sidechain – is it always intact?

L189 Surely the conformation of the intermediate is different from the lactam?

As noted above, I'd be careful about describing these structures as intermediates. The hydrolysed products could be binding – the carbapenems will be hydrolysed on the timescale of the incubation.

Why should some structures give delta-1 and some delta-2? Why not mixtures?

Discuss loop movements with regard to other MBLs.

The refinement of a deprotonated lactam derived N needs to be clearly made.

The lack of bridging water observed crystallographically with the carbapenems is interesting, but does not preclude its occurrence in solution – perhaps enabled by other changes, e.g. zinc ion movement (was there any evidence for zinc movement?). Anyway the key issue of the protonation agent in carbapenem catalysis is unclear.

L268 two possible epimers/stereoisomers (not enantiomers).

L272 by what technique – in solutions or crystallographically?

L275 Was the stereochemistry validated by NMR – it should be. Are the J values consistent?

Note carbanions can be sp^3 or sp^2 .

Figure 5 – check carefully, especially compare with ref 10 with regard to protonation states/steps.

Discussion

Will need modifying in the light of the comments above – the key think is demonstrating that the crystallographically observed complexes are representative of intermediates in solution.

Overall the discussion could be much shorter and more concise, focusing on the new insights.

Reviewers' comments:

Reviewer #2 (Remarks to the Author):

This manuscript details the experimental confirmation of a novel mechanism of carbapenem hydrolysis.

On the whole this is a very interesting manuscript, with a good level of detail, and sound evidence for the conclusions drawn.

The crystallography is sound.

158-159 suggest that the carbon is sp³ hybridised, what was the temperature factor for this atom compared to others in the ligand?

What are the confidence limits for the ligand modelling?

When comparing ligand overlay from different structure RMSD values (max and min) should be given for the reader to assess how this has been justified.

Line 161 - 164 "density fitting was improved" How? Please give the refinement statistics used to decide this.

I recommend publication with some minor corrections additions (see below).

line 185 give rmsd similarly 231.

There are a few grammatical/typographic errors e.g.

line 45 penicillins?

line 144 complexed?

and others, please check proof.

Minor additions/corrections suggested:

I think the addition of the following two papers

Ma et al. *Angew Chem Intn Edn Engl*, 2014, 53, 2130 (real time monitoring of meropenem hydrolysis by ¹H NMR)

and Groundwater, et al., *Fut. Med. Chem.*, 2016, 8, 993 (New Delhi Metallo- β -lactamase (NDM-1): Structure, Inhibitors, and Detection of Producers) is warranted.

Question: Can this mechanism potentially be extrapolated to other NDM-X variants? A brief discussion on this is warranted.

Would the authors include a comparison hydrolysis mechanism diagram for the other classes of antibiotics - even in SI would be acceptable) this would make this paper a fantastic 'one-stop-shop' for mechanism.

There does appear to be the formation of a single diastereoisomer, suggesting the stereospecific delivery of the proton from a water molecule from above. I would like to have seen the HMQC spectrum (to show ¹H - ¹³C correlations and confirm the identity of H-2) and the full ROESY spectrum

in the supplementary files. The ^{13}C spectrum in the supplementary file looks much cleaner than both the ^1H and ROESY spectra. The section on C2 enantiomers should discuss epimers, which would be the correct term.

Figure 2 in the SI was very unclear to this reader.

The authors are to be congratulated on what is a very interesting, and no doubt soon to be highly cited paper.

Reviewer #3 (Remarks to the Author):

The manuscript involves a carefully conducted crystallographic study of the NDM-1 catalyzed hydrolysis of the carbapenems imipenem and meropenem. The conclusions, based on the absence of a bridging water molecule (between the two zinc atoms), the formation of the Δ -1 unsaturated pyrroline, as well as an sp^3 hybridized carbon at C2 of the carbapenem imply that the intermediate anion is protonated from a bulk solvent molecule and not from the (absent) bridging water, as was the case with penicillins and cephalosporins. The study is done very carefully, and the conclusions may be of interest to the scientific community, and thus I recommend accepting the manuscript.

Minor correction: line 45, the word penicillins has only one e.

Responses to Reviewers' Comments

Comments from reviewer #1

Firstly, I should say that I think this work is worthy of publication and that it does indicate a different mechanism (or at least different kinetics) for MBL catalyzed carbapenem hydrolysis versus that of penicillins or cephalosporins. The new carbapenem derived complex structures are of interest – even though they are similar to those reported by the authors and other groups.

Reply: We appreciate for these positive comments.

What I thought the paper was going to describe was time-resolved studies on the reaction of carbapenems in crystals in which the crystal structures and spectroscopic studies in crystals are combined. This is not the case.

Reply: We are afraid that the reviewer has an incorrect conception in mind regarding crystallographic and spectroscopic techniques. X-ray crystallography is a technique used for determining the atomic structures of crystalline molecules including biological macromolecules such as proteins and nucleic acids. Spectroscopy is the study of the interaction between matter and electromagnetic radiation, which allows measurements of radiation intensity as a function of wavelength.

Time-resolved studies using crystallography is not a simple combination of crystallographic and spectroscopic measurements. Given the fragile nature of macromolecule crystals, spectroscopic study on crystals is technically impossible. The principle of time-resolved crystallography is to trigger a reaction of interest the crystal prior to X-ray exposure, and then collecting the diffraction patterns at different time delays. In another word, it is a technique within the scope of X-ray crystallography, and has nothing to do with spectroscopy.

Notably, time-resolved crystallography has not become a widely used technique in studying dynamical changes of enzymatic catalysis to date, due to a few experimental limitations. To our best knowledge, no study of β -lactamase-catalyzed reactions by means of time-resolved crystallography has

been reported yet.

NMR studies simply analyse the products, not intermediates.

Reply: Similar to previous studies of β -lactamase-catalyzed reactions (Mobashery and Johnston 1986, Pratt and Faraci 1986, Ma, McLeod et al. 2014), the process of β -lactam hydrolysis was monitored by ^1H NMR spectra in this study, which however impossibly reveal the stereochemical structure of the intermediates by itself. To this end, 2D NMR spectra are needed, but dissimilar to 1D proton spectra, the signals recorded in 2D spectra such as ^1H - ^1H COSY, ROESY and ^1H - ^{13}C HSQC etc. is very weak such that a complete 2D spectrum has to be collected in much longer time windows, usually in scale of hours or even days. It means that hydrolytic intermediate structure determination by means of NMR spectroscopy is in principle impossible due to fast turnover rates of β -lactam hydrolysis, which would end in several minutes.

Fortunately, intermediate structures were well resolved in our crystals. The role of NMR spectroscopy in this study was to confirm if β -diastereomeric product was solely generated in carbapenem hydrolysis as suggested from the crystal structures. The good convergence of our NMR and crystallographic data sufficiently support the conclusion that hydrolyzed carbapenems are likely protonated by a donor different from that in penicillin and cephalosporin hydrolysis, which has been acknowledged by the other two reviewers.

The crystallography involves soaking – hence it cannot be ruled out that the crystallographically observed complexes arise from hydrolysed products (re)binding.

Reply: Other than cocrystallization of enzyme and substrate, crystal soaking with substrates is another commonly used approach for trapping reaction intermediates. The rationale of crystal soaking is based on a dogma that enzymatic reactions are properly proceeded in crystals but with slower turnover rates than in solution by several orders of magnitude. Such a conception has been proven by independent studies performed in early era of X-ray

crystallography (1960~70s), e.g. the reports concerning carboxypeptidase A (Lipscomb 1973, Spilburg, Bethune et al. 1977), and now is generally believed by crystallographers and biochemists.

In previous crystallographic studies of β -lactam hydrolysis, a number of cocrystals containing enzyme-intermediate/product adducts, in particular those containing carbapenem intermediates, were obtained by crystal soaking, e.g. OXA-1 in complex with doripenem (Schneider, Karpen et al. 2009) and GES-1/GES-5 in complex with imipenem (Smith, Frase et al. 2012). In the current study, we at first tried cocrystallization of NDM-1 with imipenem/meropenem, but failed to see density corresponding to lactam molecules, indicating that (i) carbapenems were completely hydrolyzed before the enzyme molecules had packed in crystal lattice, and (ii) hydrolyzed carbapenems have much lower affinity to NDM-1 than hydrolyzed ampicillin or cephalosporin antibiotics, which could be cocrystallized with NDM-1 (Zhang and Hao 2011, King, Worrall et al. 2012, Kim, Cunningham et al. 2013, Feng, Ding et al. 2014).

We then used crystal soaking in our experiment and successfully obtained complex crystals, in which three enzyme-intermediate/product adducts, EI₁, EI₂ and EP, were trapped. High occupancy (0.7 ~ 1.0) of these reaction species after refinement sufficiently indicates that the substrate can permeate into crystal lattice and be hydrolyzed in crystal milieu, while product release from the enzyme is effectively restrained by crystal lattice. Consequently, the chance for hydrolyzed product rebinding, even not completely impossible, is extremely low, as most active sites in crystallized enzyme have been occupied by intermediates or unreleased products generated from in situ hydrolysis.

I'm also unconvinced that at the resolution described the quality of the maps is sufficient to unequivocally define ratios of data 1/2 isomers – only to predict which is likely to predominate.

Reply: Our crystal structures were determined around 2 Å resolution. The electron density maps generated at such resolution are essentially good enough to unequivocally distinguish sp² and sp³ hybridization of a certain atom, as these

two states correspond to planar and puckered density respectively, which is quite easy to be distinguished. A convincing example was the crystal structure of NDM-1 in complex with hydrolyzed cephalixin (PDB 4RL2) recently determined at 2.0 Å in our laboratory, which clearly displayed formation of a chiral center (sp^3 hybridization) of C3 in cephalixin intermediate (Feng, Ding et al. 2014). With comparable quality to that structure, sp^2 and sp^3 hybridization of the C2 atom in imipenem/meropenem, i.e. the Δ^2 and Δ^1 isomers, were undoubtedly defined in our structures determined in this study as well, as shown in Figure 2.

It is necessary to mention that transition intermediates with delocalized negative charge are impossibly trapped in a crystal, because such intermediates between Δ^2 and Δ^1 are highly unstable and transient.

One option would be for the authors to publish the results in a specialist crystallography journal (which I'm supportive of).

Reply: Sorry, we do not think that this suggestion is an acceptable idea because crystallography journals principally publish novel protein or nucleic acid structures, but the NDM-1 structure is not such a structure. As a matter of fact, the structures of NDM-1 alone and in complex with various substrates or inhibitors have been published in a number of articles since 2011 (Green, Verma et al. 2011, Zhang and Hao 2011, King, Worrall et al. 2012, Kim, Cunningham et al. 2013, Brem, Cain et al. 2016, Hinchliffe, Gonzalez et al. 2016), even including an article from our own laboratory (Feng, Ding et al. 2014). The majority of these articles reported more mechanistic than structural findings and were published on chemistry/biochemistry oriented or comprehensive journals rather than crystallography-specific journals.

Alternatively, they should carry out spectroscopic studies (UV/vis and, maybe, IR/Raman) on the same crystals on which they collect time-resolved diffraction data in order to validate the density they observe as intermediates.

Reply: We have discussed with spectroscopy experts regarding the possibility of this request, and all of them thought it is technically impossible. Almost all

spectroscopic experiments including UV/vis, Raman and infrared spectroscopies are carried out on samples in solution, although in a few cases of solid state spectroscopic studies are performed on samples prepared in powders or immobilized on membranes. Apparently, protein crystals cannot be treated in either of these ways, which would definitely destroy crystal lattice and even the 3D structure of the destination molecules.

They should also rule out the possibility of products rebinding to the enzyme.

Reply: This request is either unnecessary or technically infeasible. As we have explained in above answers, there is little chance for the hydrolyzed product in solution to rebind to the enzyme within crystal lattice, which has been occupied by reaction intermediate or unreleased product resulting from hydrolysis proceeded in crystal milieu. In practical terms, there lacks an experimentally feasible way to effectively restrain product rebinding but not block substrate binding since both compounds can bind to the enzyme probably without significant difference in terms of binding affinity.

L38 add citation.

Reply: A citation has been added.

L62 NDM-1 is not the most recent MBL – NDMs may be the most recent additional family – check.

Reply: We agree to this comment and have used NDMs to replace NDM-1 in the corresponding paragraph.

L67 some SBLs do catalyse carbapenem hydrolysis. L68 'are resistant to many SBLs'.

Reply: We are sorry for our imprecise expression of “resistant to most serine β -lactamases”, which was modified to “resistant to many serine β -lactamases” in the new version of manuscript.

L 76-77 use standard BBL numbering throughout

Reply: We do not think it is wise to follow this suggestion. This manuscript is a report solely on NDM-1. Dissimilar to review articles, authors of such a research paper usually do not use BBL numbering in order to avoid confusions for readers who are not familiar with the MBL family. In conformity to this convention, specific numbering for the enzyme in question, for example D124 rather than D120, was used in all previously published research articles of NDM-1 (King and Strynadka 2011, Zhang and Hao 2011, King, Worrall et al. 2012, Feng, Ding et al. 2014, Tripathi and Nair 2015).

L82 'proceed in two stages'?

Reply: As is well reviewed in a number of references (Palzkill 2013, Smith, Antunes et al. 2013, Meini, Llarrull et al. 2015), MBL-mediated β -lactam hydrolysis proceeds via two steps: cleavage of the amide bond and decay of the anionic hydrolytic intermediate. This hydrolyzing pathway has been generally accepted by scholars doing research in the relevant field.

L86 how can the anion be generated whilst the β -lactam is intact?

Reply: We do not understand why the reviewer thought that the anion might be generated from intact β -lactam, which is beyond common senses with regard to a hydrolytic reaction. We think that our statement "Synchronously with the opening of the ring, an anionic intermediate is generated..." appropriately and clearly describes the process of anion generation occurring in concert with the cleavage of the C-N bond. More straightforwardly speaking, the anion generated whilst the β -lactam is opened.

L91 whether anion intermediate protonation is rate limiting should depend on the substrate?

Reply: This is a good question that has not been unequivocally addressed. Although most researchers have accepted the idea that protonation of the anion intermediate is the rate-limiting step in the process of β -lactam hydrolysis, solid experimental evidence firmly linking this step and substrate composition is unfortunately lacking. Actually, the data shown in our manuscript strongly

suggest that the proton donor is substrate-dependent, which probably prompts that the rate of intermediate protonation may also depend on the substrate.

L104 add citation.

Reply: We do not think it is necessary to do so because a citation was already added to the next sentence following L104 in the current manuscript.

L106 the crystallography does inform in the protonation state at N1 or is this inferred?

Reply: A double bond between C4 and N5 (N1 in the reviewer's question) has been clearly revealed in crystal structures of PDB 4rl0 and 4rl2 (Feng, Ding et al. 2014), which unquestionably exclude the possibility of protonation of N5 due to the sp² hybridization of this nitrogen atom.

L110-112 there are other reports, see e.g. Chem Sci 2015, 6, 956-963.

Reply: To our best knowledge, ref. 25 (Tioni, Llarrull et al. 2008) is the only available publication so far reporting possible double bond tautomerization in carbapenems hydrolyzed by an MBL, as two diastereomeric products were detected by Raman spectroscopy. The reference mentioned by the reviewer, however, is a report concerning biophysical and kinetic studies on SPM-1, another B1 MBL subfamily member. An active-site loop movement revealed by crystallography and NMR spectra was the major finding in that study, which yet did not detect any β -lactam intermediates or products. Therefore, we do not think it is appropriate to cite that publication here.

L134 note the tautomerization has been previously addressed – see ref 10.

Reply: Ref. 10 is a review article that discusses double bond tautomerization in MBL-catalyzed carbapenem hydrolysis by citing the experimental results reported in ref. 25 (Tioni, Llarrull et al. 2008). Although that study has inferred the occurrence of tautomerization by the detection of two diastereomeric products, direct observation of carbapenem tautomers has never been obtained

from MBL-mediated hydrolysis. By contrast, both Δ^2 and Δ^1 tautomers have been revealed in a number of SBL-carbapenem complex structures. In this context, an important aim of our study was to explore if pyrroline tautomerization could be directly observed in crystal structures with imipenem/meropenem bound at the active site of NDM-1.

L143-153 Giving soaking times / enzyme /lactam ratios.

Reply: All details of crystal soaking, including soaking time and protein-antibiotics molar ratios, were already given in the subsection of “Crystallization and diffraction data collection” of Experimental Procedures (L422-426) and also in Supplementary Information in the current manuscript.

L189-192 Evidence for the EP derivative needs to be supplied.

Reply: This is a confusing request since the EP derivative is not described at all in L189-192.

L287-288 Isn't it possible the structures are derived from products (re)binding to the enzyme, in which case one of the stereoisomers may bind more tightly, biasing the results.

Reply: As we have addressed in an above reply to the same question, the possibility of product rebinding is so little to be ignored, since the active sites of NDM-1 have been highly occupied by imipenem/meropenem intermediates or unreleased products in the complex crystals.

L335-340 the key novel aspect is the stereo selective protonation – but this needs to be validated further, at least in terms of the crystallographic data.

Reply: We think that this is another both unnecessary and technically impossible request. Firstly, this key evidence in drawing our conclusion has been fully validated by the data presented in our manuscript. The single epimer corresponding to *S* at C2 observed in all Δ^2 intermediate copies determined in our crystals and the exclusive β -diastereomeric product detected in NMR spectra

provided sufficient and very solid evidence to validate the stereo-selectivity of intermediate protonation. Secondly, as we stated in Discussion (L370-373), protonation states of a certain atom are impossibly defined by crystallographic data because hydrogen atoms are invisible in X-ray structures unless in extreme cases with extraordinarily high resolution, say 0.5 Å or so.

Show mFo-Dfc map in figure 2 SI (not, just, 2Fo-fc map).

Reply: We are afraid this is another confusing request. Figure 2 in SI displays superimposition of NDM-1 structures in complex with imipenem, meropenem, ampicillin and cephalexin. No electron density map including $2F_o-F_c$ map, was shown in that figure.

L143 I do think soaking with hydrolysed carbapenems should be tried.

Reply: This is an experimentally impractical request. Hydrolyzed carbapenems are not commercially available. Moreover, once carbapenems are hydrolyzed, they usually become less stable in solution than the intact β -lactam structures (unpublished NMR data from our laboratory), making product purification from hydrolyzing reactions impossible.

L144 complex.

Reply: We feel sorry again for this confusing question. The word “complex” does not appear in L144.

L176 how do the structures compare with the reported structure – in ref 18.

Reply: The hydrolyzed meropenem present in the reported structure of ref. 18 (King, Worrall et al. 2012) shows low occupancy such that the R2 side chain is completely invisible in electron density. Even so, we did superimposition of imipenem in an EP complex and meropenem in an E1₂ complex with that structure (Fig. 3e, f), which allowed us to reason that the structure in ref. 18 more likely represent an EP complex of meropenem bound to NDM-1.

Comment on the hydroxyethyl sidechain – is it always intact?

Reply: The hydroxyethyl side chain in both imipenem and meropenem is intact in all antibiotics copies present in our structures.

L189 Surely the conformation of the intermediate is different from the lactam?

Reply: It is impossible to know the conformational difference between the intermediate and the antibiotics with unopened lactam, since no MBL structure in complex with an intact substrate is available to date.

As noted above, I'd be careful about describing these structures as intermediates. The hydrolysed products could be binding – the carbapenems will be hydrolysed on the timescale of the incubation.

Reply: We have answered the unrealistic “concern” about product (re)binding twice. Please refer to the above replies.

Why should some structures give delta-1 and some delta-2? Why not mixtures?

Reply: We do not know the exact meaning of “mixture” from the reviewer’s question. If it means a transition state between Δ^2 and Δ^1 , we can definitely say that trapping such a state in crystals is impossible, because an intermediate containing delocalized electron is in principle highly unstable and very transient. If it means presence of both Δ^2 and Δ^1 in a single crystal, we think it is possible. In fact, intermediates generated at different reaction steps have been trapped in different enzyme subunits in quite a number of crystal structures. In this study, however, all carbapenem copies present in any a single crystal displayed very good consistence in pyrroline tautomerization. We think this is attributable to the strict non-crystallographic symmetry among NDM-1 copies in the asymmetric unit, which gives identical chemical microenvironment to each enzyme molecule.

Discuss loop movements with regard to other MBLs.

Reply: Loop movement upon substrate binding in particular the comparison with other MBLs is out of the topic of our manuscript. Even so, we described the

inconsistent loop conformations of L3 in NDM-1 between imipenem and meropenem bound structures with ampicillin and cephalexin at L195 -200 and Supplementary Fig. 2.

The refinement of a deprotonated lactam derived N needs to be clearly made.

Reply: The details of ligand modeling and refinement are given in the subsection “Structure determination and refinement” of Experimental Procedures. In brief, models of hydrolyzed imipenem/meropenem in either Δ^2 and Δ^1 tautomers are manually built using the ligand builder in *Coot*, and the corresponding topological dictionaries were generated from the *ProDRG2* server. For Δ^1 tautomers, the lactam nitrogen is automatically assigned in deprotonated state due to the sp^2 hybridization, while for Δ^2 tautomers, the dictionary file was manually edited to force it to be negatively charged with a valence of -1.

The lack of bridging water observed crystallographically with the carbapenems is interesting, but does not preclude its occurrence in solution – perhaps enabled by other changes, e.g. zinc ion movement (was there any evidence for zinc movement?). Anyway the key issue of the protonation agent in carbapenem catalysis is unclear.

Reply: We are afraid that the reviewer may not fully comprehend our interpretation about the missing water bridging the two zinc ions. Structure comparisons shown in Figure 3g-h have unquestionably attribute the absence of this crucial solvent molecule to the unique orientation of the newly formed C6-carboxylate structurally determined by the trans configuration of the C5-C6 bond in carbapenem lactam (Fig. 1). Consequently, steric hindrance is unavoidably formed between one oxygen atom in C6-carboxylate with a water molecule presumably sitting between the zinc ions. It means that exclusion of the bridging water is solely determined by the unique stereochemical structure of carbapenems and is therefore completely independent of whether they are hydrolyzed in crystals or in solution or how zinc ion moves during the hydrolysis, which are confined to 3.8 ~ 4.6 Å as reported in previously published structures.

L268 two possible epimers/stereoisomers (not enantiomers).

Reply: We would like to thank the reviewer to point out this terminology misuse. The word “epimer” was used in place of enantiomer in the revised manuscript.

L272 by what technique – in solutions or crystallographically?

Reply: As described in ref. 25 (Tioni, Llarrull et al. 2008), two diastereomeric products from BcII-mediated hydrolysis of imipenem were detected using Raman spectroscopy with the reaction system in solution.

L275 Was the stereochemistry validated by NMR – it should be. Are the J values consistent?

Reply: We did not measure J values, but instead, multiple 2D spectra such as ^1H - ^1H DEPT, COSY, TOCSY, ROESY, and ^1H - ^{13}C HSQC, HMBC were collected in this study. These 2D spectra, in particular ^1H - ^1H ROESY (shown in Fig. 4b), provided straightforward and more convincing evidences than J values in validation of the stereochemistry of meropenem-hydrolyzed products.

Note carbanions can be sp^3 or sp^2 .

Reply: Since the reviewer did not give the corresponding line number, we cannot figure out what he meant. In principle of organic chemistry, carbanions can be either sp^3 or sp^2 , but for β -lactam intermediates only sp^3 carbanions in the pyrroline ring may exist after double bond tautomerization.

Figure 5 – check carefully, especially compare with ref 10 with regard to protonation states/steps.

Reply: This is another confusing request to us. Figure 5 shows a potential carbapenem-hydrolyzing pathway reacted by NDM-1, which was reasoned out solely from our experimental data presented in our manuscript. We cannot see any necessity to compare this figure with ref. 10 (Brem, Cain et al. 2016), although these two figures agree each other in many details such as negative charge delocalization and generation of carbanionic intermediates. Even so, there are some crucial differences between them. A branched mechanism was

proposed in Figure 2B of ref. 10, while a linear reaction mechanism was summarized in Figure 5 of our manuscript as only β -diastereomeric product was detected in this study. Given the solid structural data obtained in this study, which was however lacking in ref. 10, we think that our figure is an advanced version depicting the potential carbapenem-hydrolyzing mechanism reacted by MBLs.

Discussion will need modifying in the light of the comments above – the key think is demonstrating that the crystallographically observed complexes are representative of intermediates in solution.

Reply: We do not agree to modify the Discussion according to the comments raised by the reviewer. As is clearly shown in above answers, we believe most of these comments are based on incorrect understanding of crystallographic and spectroscopic techniques, and insufficient comprehension of our experimental data obtained in this study (e.g. the structural determinant for the absence of the bridging water and the ignorable chance of product rebinding to enzymes in crystals). Given the good convergence of our crystallographic and NMR spectroscopic data presented in our manuscript, we insist that the enzyme-intermediate or enzyme-product adducts revealed in our crystals well represents the reaction pathway in solution.

Comments from reviewer #2

This manuscript details the experimental confirmation of a novel mechanism of carbapenem hydrolysis.

On the whole this is a very interesting manuscript, with a good level of detail, and sound evidence for the conclusions drawn.

The crystallography is sound.

Reply: We appreciate for all of these positive comments.

158-159 suggest that the carbon is sp^3 hybridised, what was the temperature

factor for this atom compared to others in the ligand?

Reply: In our structures, the sp³-hybridized atom (C2) shows comparable temperature factor with the neighboring atoms in the pyrroline ring, which is significantly lower than the bonded sulfur atom. For examples, in all five imipenem-bound structures representing EI₂ or EP complex, the temperature factors of C2, C1, C3 and the sulfur are 27.13 ~ 48.23, 27.03 ~ 48.40, 27.25 ~ 48.47 and 41.97 ~ 67.44, respectively. This temperature factor profile reflects good rigidity of the pyrroline ring and high mobility of the R2 side chain, which in good consistence with the previously published carbapenem-bound structures of SBLs.

What are the confidence limits for the ligand modelling?

Reply: From crystallographic point view, the confidence limits for ligand modeling comes from ligand-density fitting, which may be reflected by refinement statistics in R-factors, occupancies and temperature factors, but is mostly checked by eyes. As we explained above to a question from Reviewer #1, the resolution around 2.0 Å for our crystal structures allow us to model ligands with high confidence, as the electron density maps generated at such resolution are good enough to unequivocally distinguish sp² and sp³ hybridization of a certain atom, which correspond to planar and puckered density respectively. With comparable quality to the recently determined 2.0 Å structure of NDM-1 in complex with hydrolyzed cephalixin (PDB 4RL2) by our laboratory (Feng, Ding et al. 2014), sp² and sp³ hybridization of the C2 atom in imipenem/meropenem, i.e. the Δ² and Δ¹ isomers, were well defined in our structures described in this manuscript.

When comparing ligand overlay from different structure RMSD values (max and min) should be given for the reader to assess how was this has been justified.

Reply: We agree that RMSD values should be given when two structures are compared. However, there is a practically tricky problem in calculating RMSDs between overlaid ligands. Dissimilar to macromolecules such as proteins or

nucleic acids, the RMSDs may not reflect the overlaying degree for small molecules. Flexible groups in compounds may lead to large RMSDs even if the overall structure can be well superposed. Unfortunately, hydrolyzed imipenem belongs to such a class in that it contains a highly mobile R2 side chain. Consequently, although the structural core of imipenem (hydrolyzed lactam and the pyrrolidine ring) in Δ^2 and Δ^1 is perfectly overlaid to each other (Fig. 2f), the RMSDs between them reached to 1.02 ~ 1.61 Å. By contrast, overlays of hydrolyzed meropenems containing a more rigid R2 side chain gives RMSDs of 0.29 ~ 0.83 Å between the two isomers. More strikingly, overlays between these two carbapenem substrates lead to RMSDs even above 3.0 Å, significantly deviating from values reasonably reflecting the well-overlaid structural cores (Fig. 2g and 3e). Given large RMSD fluctuations contributed by side chain flexibility, we do not think it is wise to provide these values in the manuscript, which may mislead the readers.

Line 161 - 164 "density fitting was improved" How? Please give the refinement statistics used to decide this.

Reply: As we have said in the manuscript, correct ligand modeling resulted in better density fitting after full-model refinement. This was mostly checked by eyes rather than by comparing the refinement statistics, because alteration between the two tautomers in carbapenem molecules leads to very tiny changes in refinement statistics that are completely negligible. The following are two examples that well illustrate the correctness of the modeled ligands.

The left figure shows an incorrect modeling of hydrolyzed imipenem in Δ^2 isomer where poor density fitting, in particular for the sulfur atom bonded to sp^2 hybridized C2, was clearly seen. By sharp contrast, the right figure shows a correct modeling of this molecule, which results in much better density fitting of C2 and the sulfur. Notably, the density surrounding C2 becomes more puckered after refinement than that with wrong modeling. Such a co-improvement of model and density is a good indicator for correct modeling of a Δ^1 tautomer in this case.

Another example of hydrolyzed meropenem modeling contrary to the above case is shown in these figures, where wrong modeling of Δ^1 leads to uncomfortable positioning of sp (left figure) although the fitting of the sulfur atom is reluctantly acceptable. In contrast, modeling of meropenem in Δ^2 significantly improves density fitting for both C2 and the sulfur (right figure). Again, the density here is co-improved and becomes more planar for better accommodation of sp^3 -hybridized C2 after refinement, which gave us more confidence in modeling of a Δ^2 isomer in this structure.

line 185 give rmsd similarly 231.

Reply: Please refer to the above explanation responding to a comment about the RMSD values for ligand overlay.

There are a few grammatical/typographic errors e.g. line 45 penicillins? line 144 complexed? and others, please check proof.

Reply: We really appreciate for the reviewer's picking up these typos. We have carefully checked our manuscript and done our best to correct spelling and grammatical errors.

Minor additions/corrections suggested: I think the addition of the following two papers. Ma et al. Angew Chem Intn Edn Engl, 2014, 53, 2130 (real time monitoring of meropenem hydrolysis by 1H NMR) and Groundwater, et al., Fut. Med. Chem., 2016, 8, 993 (New Delhi Metallo- β -lactamase (NDM-1): Structure, Inhibitors, and Detection of Producers) is warranted.

Reply: The above references and a very recent publication --- Khan et al., Structure, Genetics and Worldwide Spread of New Delhi Metallo- β -lactamase (NDM): a threat to public health BMC Microbiology 2017, 17:101 were added in the updated version of our manuscript.

Question: Can this mechanism potentially be extrapolated to other NDM-X variants? A brief discussion on this is warranted.

Reply: We believe this is an interesting proposal that was overlooked in our manuscript. Since the first characterization of NDM-1, 16 variants displaying comparable carbapenemase activities have been identified in this family. These variants differ from NDM-1 by one or two amino acid substitutions, none of which however are in the key catalytic residues, or those involved with substrate interaction or the maintenance of the active site conformation (Groundwater, Xu et al. 2016, Khan, Maryam et al. 2017). Therefore, we reason that the mechanism proposed on the basis of our crystallographic and NMR data can be extrapolated to other NDM-x variants. In response to the reviewer's suggestion, a brief discussion has been added in Discussion.

Would the authors include a comparison hydrolysis mechanism diagram for the other classes of antibiotics - even in SI would be acceptable) this would make this paper a fantastic 'one-stop-shop' for mechanism.

Reply: We would like to thank for this suggestion, which may make the mechanistic differences among β -lactam substrates clearer for readers. However,

we have two concerns in following this proposal. First, the proton donor of penicillin and cephalosporin intermediates has not been fully identified thus far. Although the bridging water may serve as an ideal proton donor for these antibiotics, the candidate list also includes the newly formed carboxylate, apical water bound to Zn² and an incoming solvent molecule (Zhang and Hao 2011, Kim, Cunningham et al. 2013, Feng, Ding et al. 2014). In another word, the protonation mechanism for penicillin and cephalosporin hydrolysis is still a matter of debate, and thus it is impossible at this time to show a definite reaction pathways for β -lactam other than carbapenems, which can be generally accepted by researchers working in this field. Second, we are not among the laboratories determining the crystal structures of NDM-1 in complex with penicillins. Considering that this manuscript is submitted as a research article rather than a review paper, we are afraid it might be inappropriate to show a figure based on data not from our own study.

There does appear to be the formation of a single diastereoisomer, suggesting the stereospecific delivery of the proton from a water molecule from above. I would like to have seen the HMQC spectrum (to show ¹H – ¹³C correlations and confirm the identity of H-2) and the full ROESY spectrum in the supplementary files.

Reply: According to the reviewer's request, we will submit the ¹H-¹³C HMBC and HSQC spectra and full ¹H-¹H ROESY as supplementary files.

The ¹³C spectrum in the supplementary file looks much cleaner than both the ¹H and ROESY spectra. The section on C2 enantiomers should discuss epimers, which would be the correct term.

Reply: Thank the reviewer for accepting our ¹³C NMR spectrum, and for pointing out the term misuse, which was also indicated by Reviewer #1. All misused words were corrected.

Figure 2 in the SI was very unclear to this reader.

Reply: We are sorry for this possibly problematic figure, although we do not have a clear idea about the reviewer's concern. Since the figure in question

shows the comparison of four key amino acids, F70, W93, K211 and N220, involved in substrate-binding, we thought that the hydrolyzed β -lactams bound at the active site may not need being highlighted by stick-and-ball models. For clarity, we modified this figure by representing the antibiotics with line models in the new version of Supporting Information.

The authors are to be congratulated on what is a very interesting, and no doubt soon to be highly cited paper.

Reply: Thank the reviewer so much for his complimentary remarks, which encourages us to do more research concerning MBL-mediated hydrolyses.

Comments from reviewer #3

The manuscript involves a carefully conducted crystallographic study of the NDM-1 catalyzed hydrolysis of the carbapenems imipenem and meropenem. The conclusions, based on the absence of a bridging water molecule (between the two zinc atoms), the formation of the Delta-1 unsaturated pyrroline, as well as an sp³ hybridized carbon at C2 of the carbapenem imply that the intermediate anion is protonated from a bulk solvent molecule and not from the (absent) bridging water, as was the case with penicillins and cephalosporins. The study is done very carefully, and the conclusions may be of interest to the scientific community, and thus I recommend accepting the manuscript.

Reply: Thank the reviewer so much for his positive comments, complimentary remarks and recommendation of manuscript acceptance! We think this is the best reward for our work.

Minor correction: line 45, the word penicillins has only one e.

Reply: Thanks for pointing out this typo. We have carefully checked our manuscript and done our best to correct spelling and grammatical errors.

References:

- Brem, J., R. Cain, S. Cahill, M. A. McDonough, I. J. Clifton, J. C. Jimenez-Castellanos, M. B. Avison, J. Spencer, C. W. Fishwick and C. J. Schofield (2016). "Structural basis of metallo-beta-lactamase, serine-beta-lactamase and penicillin-binding protein inhibition by cyclic boronates." Nat Commun **7**: 12406.
- Feng, H., J. Ding, D. Zhu, X. Liu, X. Xu, Y. Zhang, S. Zang, D. C. Wang and W. Liu (2014). "Structural and mechanistic insights into NDM-1 catalyzed hydrolysis of cephalosporins." J Am Chem Soc **136**(42): 14694-14697.
- Green, V. L., A. Verma, R. J. Owens, S. E. Phillips and S. B. Carr (2011). "Structure of New Delhi metallo-beta-lactamase 1 (NDM-1)." Acta Crystallogr Sect F Struct Biol Cryst Commun **67**(Pt 10): 1160-1164.
- Groundwater, P. W., S. Xu, F. Lai, L. Varadi, J. Tan, J. D. Perry and D. E. Hibbs (2016). "New Delhi metallo-beta-lactamase-1: structure, inhibitors and detection of producers." Future Med Chem **8**(9): 993-1012.
- Hinchliffe, P., M. M. Gonzalez, M. F. Mojica, J. M. Gonzalez, V. Castillo, C. Saiz, M. Kosmopoulou, C. L. Tooke, L. I. Llarrull, G. Mahler, R. A. Bonomo, A. J. Vila and J. Spencer (2016). "Cross-class metallo-beta-lactamase inhibition by bisthiazolidines reveals multiple binding modes." Proc Natl Acad Sci U S A **113**(26): E3745-3754.
- Khan, A. U., L. Maryam and R. Zarrilli (2017). "Structure, Genetics and Worldwide Spread of New Delhi Metallo-beta-lactamase (NDM): a threat to public health." BMC Microbiol **17**(1): 101.
- Kim, Y., M. A. Cunningham, J. Mire, C. Tesar, J. Sacchettini and A. Joachimiak (2013). "NDM-1, the ultimate promiscuous enzyme: substrate recognition and catalytic mechanism." FASEB J **27**(5): 1917-1927.
- King, D. and N. Strynadka (2011). "Crystal structure of New Delhi metallo-beta-lactamase reveals molecular basis for antibiotic resistance." Protein Sci **20**(9): 1484-1491.
- King, D. T., L. J. Worrall, R. Gruninger and N. C. Strynadka (2012). "New Delhi metallo-beta-lactamase: structural insights into beta-lactam recognition and inhibition." J Am Chem Soc **134**(28): 11362-11365.
- Lipscomb, W. N. (1973). "Enzymatic activities of carboxypeptidase A's in solution and in crystals." Proc Natl Acad Sci U S A **70**(12): 3797-3801.
- Ma, J., S. McLeod, K. MacCormack, S. Sriram, N. Gao, A. L. Breeze and J. Hu (2014). "Real-time monitoring of New Delhi metallo-beta-lactamase activity in living bacterial cells by ¹H NMR spectroscopy." Angew Chem Int Ed Engl **53**(8): 2130-2133.

- Meini, M. R., L. I. Llarrull and A. J. Vila (2015). "Overcoming differences: The catalytic mechanism of metallo-beta-lactamases." FEBS Lett **589**(22): 3419-3432.
- Mobashery, S. and M. Johnston (1986). "Reactions of Escherichia coli TEM beta-lactamase with cephalothin and with C10-dipeptidyl cephalosporin esters." J Biol Chem **261**(17): 7879-7887.
- Palzkill, T. (2013). "Metallo-beta-lactamase structure and function." Ann N Y Acad Sci **1277**: 91-104.
- Pratt, R. F. and W. S. Faraci (1986). "Direct observation by proton NMR of cephalosporate intermediates in aqueous solution during the hydrazinolysis and beta-lactamase-catalyzed hydrolysis of cephalosporins with 3' leaving groups: kinetics and equilibria of the 3' elimination reaction." J Am Chem Soc **108**(17): 5328-5333.
- Schneider, K. D., M. E. Karpen, R. A. Bonomo, D. A. Leonard and R. A. Powers (2009). "The 1.4 Å crystal structure of the class D beta-lactamase OXA-1 complexed with doripenem." Biochemistry **48**(50): 11840-11847.
- Smith, C. A., N. T. Antunes, N. K. Stewart, M. Toth, M. Kumarasiri, M. Chang, S. Mobashery and S. B. Vakulenko (2013). "Structural basis for carbapenemase activity of the OXA-23 beta-lactamase from Acinetobacter baumannii." Chem Biol **20**(9): 1107-1115.
- Smith, C. A., H. Frase, M. Toth, M. Kumarasiri, K. Wiafe, J. Munoz, S. Mobashery and S. B. Vakulenko (2012). "Structural basis for progression toward the carbapenemase activity in the GES family of beta-lactamases." J Am Chem Soc **134**(48): 19512-19515.
- Spilburg, C. A., J. L. Bethune and B. L. Valee (1977). "Kinetic properties of crystalline enzymes. Carboxypeptidase A." Biochemistry **16**(6): 1142-1150.
- Tioni, M. F., L. I. Llarrull, A. A. Poeylaut-Palena, M. A. Marti, M. Saggi, G. R. Periyannan, E. G. Mata, B. Bennett, D. H. Murgida and A. J. Vila (2008). "Trapping and characterization of a reaction intermediate in carbapenem hydrolysis by B. cereus metallo-beta-lactamase." J Am Chem Soc **130**(47): 15852-15863.
- Tripathi, R. and N. N. Nair (2015). "Mechanism of meropenem hydrolysis by New Delhi Metallo-beta-lactamase." Acs Catal **5**: 2577-2586.
- Zhang, H. and Q. Hao (2011). "Crystal structure of NDM-1 reveals a common beta-lactam hydrolysis mechanism." FASEB J **25**(8): 2574-2582.

Reviewers' comments:

Reviewer #1 (Remarks to the Author):

First I should say that I do think the new carbapenem derived NDM-1 structures are worthy of publication (though probably in a more specialist journal. However, I do not think the paper should be accepted as it is currently written and not without more experiments. It may be acceptable with substantial new experiments (and very considerable rewriting), but I think this would in effect be an entirely new paper.

There are many minor issues to be fixed but the major one concerns the issue as to whether the structures actually represent intermediates. It is simply not true to say spectroscopy on macromolecule crystals is impossible. For review see (e.g.) *Dynamic structural science: recent developments in time-resolved spectroscopy and X-ray crystallography*, BIOCHEMICAL SOCIETY TRANSACTIONS 2013, 41, 1260-1264. Further, the authors should do the spectroscopic characterisation even in the absence of time resolved work – the spectra will inform on the precise nature of the refined complexes, e.g. the position of the double bond (see e.g. use of Raman analysis of reacting crystals by Che et al. *JACS*, 2012, 134, 11206, though uv-Vis is the starting point).

This type of question has been extensively discussed in the literature wrt crystallographically observed structures. The authors arguments as to why not to do the spectroscopic experiments on single crystals are not convincing – several (at least) synchrotrons are now set up to do these experiments.

Crystal soaking might give structures representative of intermediates, but it might not – one way to test this is to see if the crystals react; of course the putative crystallographically observed intermediates should be correlated with solution work (and not imply product analysis as done here). I appreciate this is challenging but there is good precedent. Surely also spectroscopic work would help validate the structural assignments which in my view are not completely secure at 2Å resolution - can the authors really be sure there isn't 20% or less of another ligand bound - note there are multiple unassigned peaks in the NMR.

If the authors' arguments on soaking and intermediate are correct why don't they see binding of the intact carbapenems?

I don't see why testing for rebinding of the hydrolysed products is technically unfeasible, at least to try – see papers on carbapenem degradation – if the hydrolysed products are made and then shown not to bind by solution assays and not to be able to be soaked in under the same conditions where intact carbapenems give product type complexes, then it evidence for the products not binding and giving rise to the observed ligand structures – I'd just do the experiments!

I don't understand the reply to the NMR point – the NMR and crystallography measure different things the NMR measures the product not intermediates. Moreover at least some NMR on carbapenem hydrolysis has been published – see e.g. Smith et al. *J. Pharmaceutical Sciences*, 1990, 79, 732. (I do appreciate more modern NMR methods are used in the current paper.) Note there are many unassigned resonances in the NMR.

The main issue is that without the time resolved aspect these are 'just' further structures of an already well-characterised enzyme class; note hydrolysed carbapenem structures have been reported with MBLs, see Wachino et al. *Antimicrob. Agents Chemother.* 2016, 60, 4274.

Overall, I'm sorry not to be more positive – in my view the structures (which are of interest to

specialists especially with respect to the differences to the analogous complexes formed from penicillin and cephalosporin binding) could be reported in a specialist journal with appropriate caveats on what they represent – alternatively the authors could work to provide a more complete and convincing mechanistic picture including by doing time resolved crystallographic and solution work.

Reviewer #2 (Remarks to the Author):

I have read the authors responses to my queries, and I am satisfied that they have been addressed.

I recommend publication.

Reviewer #3 (Remarks to the Author):

This is a reasonably significant contribution to the mechanistic study of NDM-1 catalyzed hydrolysis of carbapenems.

Several grammatical errors remain:

line 62: substitute 'concern' for 'concerns'

line 67: substitute 'hydrolytic' for 'hydrolyzing'

line 85: substitute 'scaffolds' for 'natures'

line 85: substitute 'would' for 'is supposed to'

line 193: substitute 'remaining' for 'rest'

line 225: meropenem is misspelled

line 225: substitute 'consistent' for 'in good consistence'

line 226: substitute 'This' for 'It'

line 259: substitute 'configurational isomers' for 'enantiomers'

(there are additional stereogenic centers in the structure, so the term 'enantiomers' is not accurate)

line 283: substitute 'configurational' for 'enantiomeric'

line 310: substitute 'hydrolytic' for 'hydrolyzing'

line 337: substitute 'weakened' for 'weaken'

line 341: substitute 'NMR' for 'NRM'

line 351: substitute 'decipher' for 'reason out'

Responses to Reviewer #1's Comments

There are many minor issues to be fixed but the major one concerns the issue as to whether the structures actually represent intermediates. It is simply not true to say spectroscopy on macromolecule crystals is impossible. For review see (e.g.) Dynamic structural science: recent developments in time-resolved spectroscopy and X-ray crystallography, BIOCHEMICAL SOCIETY TRANSACTIONS 2013, 41, 1260-1264. Further, the authors should do the spectroscopic characterisation even in the absence of time resolved work – the spectra will inform on the precise nature of the refined complexes, e.g. the position of the double bond (see e.g. use of Raman analysis of reacting crystals by Che et al. JACS, 2012, 134, 11206, though uv-Vis is the starting point).

Reply: We agree to the reviewer's comments that spectroscopy may help to assign electron density to a particular structural or electronic conformation of crystallized complexes. Such an example was reported in an article cited by the reviewer (Che T. et al., JACS 2012, 14:11206-11215). According to that report, SA-1-204, a class D β -lactamase inhibitor, undergoes several intermediate states from W to Z during hydrolysis, with a complex mechanism involving double bond arrangement and fused ring formation. In such a case, empirical defining an intermediate structure to electron density is probably difficult, and spectroscopic measurements on crystals may be helpful in providing significant information for intermediate modeling in refined structures.

The hydrolysis of β -lactam antibiotics, however, is quite dissimilar to inhibitor hydrolysis in terms of hydrolyzing rate and mechanism. In fact, the lactamase-catalyzed hydrolysis of authentic substrates is reacted much faster with fewer intermediates generated before product release. Lots of spectroscopic studies of MBL-catalyzed carbapenem hydrolysis have been done on aqueous samples (e.g. Tioni, M. et al., JACS 2008, 130: 15852-15863), and an

intermediate containing a delocalized negative charge in the pyrroline ring was detected. On the basis of much experimental data, a generally accepted mechanism of carbapenem hydrolysis was proposed, including a transition from the Δ^2 to Δ^1 isomer (Please refer to Meini et al. *FEBS Lett* 2015, 589: 3419-3432). According to this mechanism, only two reaction intermediates, Δ^2 (EI₁) and Δ^1 (EI₂) are generated during hydrolysis, very similar to an SBL-catalyzed reaction. Nevertheless, the spectroscopic studies only provided indirect evidence for the generation of these intermediates. Our crystallographic data reported here, however, provide the direct evidences for the presence of both tautomeric intermediates in an MBL-catalyzed reaction. The modeled intermediate structures perfectly fitted into the electron density are in good consistency with those derived from spectroscopic studies, and this is why we are confident that the trapped species in our crystal structure represent the reaction intermediates.

Although we do not think it is necessary to do spectroscopic characterization for indirect evidence of double bond position, which has been clearly revealed in our crystal structures refined around 2.0 Å, the reviewer's suggestion inspires us in another respect. Spectroscopic crystallography may be a routine technique in the future that are helpful to provide more kinetic details, for example of the transition from EI₁ to EI₂ and/or from EI₂ to EP in carbapenem turnover. We are interested in such experiments once our laboratory has conditions to do that.

This type of question has been extensively discussed in the literature wrt crystallographically observed structures. The authors arguments as to why not to do the spectroscopic experiments on single crystals are not convincing – several (at least) synchrotrons are now set up to do these experiments.

Reply: We are sorry for our limited knowledge about the technical developments of time-resolved spectroscopy in recent years. By reading a review articles cited by the reviewer (Trincao J. et al., *BIOCHEM SOC T* 2013, 41:

1260-1264.), we realized that time-resolved spectroscopy has been successfully coupled to X-ray crystallography with instrumentation of UV-visible, fluorescence or Raman spectrophotometers on synchrotron beamlines. Sure, collection of time-resolved spectra on single crystals would benefit for following the kinetic procedure of carbapenem hydrolysis reacted by an MBL.

There are two limitations disallowing us to conduct the time-resolved spectroscopic measurements in this study. First, as is pointed out in the abovementioned review article (Trincao J. et al., Dynamic structural science: recent developments in time-resolved spectroscopy and X-ray crystallography, *BIOCHEM SOC T* 2013, 41: 1260-1264.), collection of spectroscopic data at synchrotron beamlines is not a routine technique. Instrumentation of UV-Vis, fluorescence or Raman spectrophotometers are only available for a few beamlines, e.g. the macromolecular crystallography beamline X10SA of the Swiss Light Source (Pompidor G. et al., *J Synchrotron Rad* 2013, 20: 765-776) and the X-ray crystallography beamline X26-C at the National Synchrotron Light Source in USA (Stoner-Ma D. et al. *J Synchrotron Rad* 2011, 18:37-40). There is no such a beamline in China, and unfortunately we do not have enough funding in this project that can support us to do such experiments in Europe or USA.

Second, spectroscopic technique is an indirect measure that requires experience and specific knowledge for spectrum interpretation. In addition, peak assignment of a spectrum may need the assistance of quantum mechanical calculations, as is described in his cited article (Che T. et al., *JACS* 2012, 14:11206-11215). Our expertise, however, concentrates in X-ray crystallography and NMR, and thus we have to cooperate with researchers who have rich experience in the field of spectroscopy and QM calculation. It is not possible for us to establish intimate cooperation with such experts in short time.

Crystal soaking might give structures representative of intermediates, but it might

not – one way to test this is to see if the crystals react; of course the putative crystallographically observed intermediates should be correlated with solution work (and not imply product analysis as done here). I appreciate this is challenging but there is good precedent. Surely also spectroscopic work would help validate the structural assignments which in my view are not completely secure at 2Å resolution - can the authors really be sure there isn't 20% or less of another ligand bound - note there are multiple unassigned peaks in the NMR.

Reply: The reviewer's opinion of "crystal soaking might or might not give rise to structures representative of intermediates" is right. However, we do not think it is necessary to test if the crystals react. The rationale of crystal soaking is based on a dogma that enzymatic reactions can properly proceed in crystals though with different kinetics from that in solution. Such a notion has been proven in countless experiments and generally believed by biochemists. If the NDM-1 crystals do not react, there will be two possible structures in the crystal: 1, the enzyme in complex with intact substrates if the crystallized enzyme remains binding affinity with substrates, or 2, the enzyme alone without any compound bound to its active site. All the 11 crystal structures determined in this study, however, revealed complex structures of NDM-1 with hydrolyzed imipenem or meropenem, undoubtedly indicating that NDM-1 hydrolyzes carbapenems in crystals simply as in aqueous solution.

As for the correlation with solution work, we have answered in above responses. In brief, our crystal structures agree well with the previously reported spectroscopic data from MBL-catalyzed reactions performed on solution samples (e.g. Tioni, M. et al., JACS 2008, 130: 15852-15863). Resolutions around 2.0 Å are high enough to unequivocally distinguish the sp² and sp³ hybridization of the C2 atom in imipenem/meropenem, which corresponds to the Δ² and Δ¹ isomeric intermediates, respectively. The difference in electron density of these two

intermediates can be clearly seen in Figure 2A – E.

In reality, the chance for incorporation of another ligand into a single crystal is extremely low and thus often thought to be impossible. According to crystallographic principles, the environment of all asymmetric units inside a crystal must be identical, which results in an overwhelming tendency for only a single compound to access into all units provided that the compound is in excess to the enzyme. In our crystal soaking trails, imipenem/meropenem in molar ratio of 5:1 to NDM-1 was added in crystallization drops, ensuring substrate-saturated incorporation into the crystal before the start of hydrolysis. Owing to the homogeneous chemical environment in the crystal, the reaction must proceed synchronously in all asymmetric units. This means that all carbapenem copies present in a certain crystal structure are supposed to represent one of a single species (EI₁, EI₂ or EP) rather than a mixed species.

The unassigned resonance signals in our NMR spectra arose from spontaneous degradation of hydrolyzed meropenem in solution, which is quite common for carbapenem and cephalosporin hydrolysis (e.g. Vilanova B. et al., J Chem Soc Pekin Trans, 1997, 2: 2439-2444). Once meropenem is hydrolyzed by an MBL, the product would continue to break down but with a very slow rate (Mendez, A. International journal of pharmaceutics 2007, 350: 97-102), which resulted in detection of trace amount of further degraded meropenem in 2D NMR spectra that were collected in 20 h. The signal/noise of these resonances, however, was not high enough for peak assignment.

If the authors' arguments on soaking and intermediate are correct why don't they see binding of the intact carbapenems?

Reply: According to the generally accepted mechanism, MBL-catalyzed β -lactam hydrolysis proceeds in two steps, cleavage of the lactam bond (from ES to EI) and

intermediate protonation (from EI to EP). The rate of the first step is much faster than that of the second step by at least one order of magnitude (e.g. Yang, H., *Biochemistry* 2012, 51: 3839-3847). It means that once an ES complex is formed at the active site of NDM-1 even in a crystal, the lactam bond is cleaved very fast, while the following step of intermediate decay is much slower. This is why complexes representing EI or EP have been trapped in a number of β -lactamase crystals, but ES never.

I don't see why testing for rebinding of the hydrolysed products is technically unfeasible, at least to try – see papers on carbapenem degradation – if the hydrolysed products are made and then shown not to bind by solution assays and not to be able to be soaked in under the same conditions where intact carbapenems give product type complexes, then it evidence for the products not binding and giving rise to the observed ligand structures – I'd just do the experiments!

Reply: This is the second time for the reviewer to request testing for product rebinding. We are sorry for not giving a clear reply in the last response letter. Actually we do not think that such a test is necessary because the binding of hydrolyzed product to the enzyme has already been confirmed in previously reported studies. In a spectroscopic study of NDM-1-catalyzed hydrolysis of nitrocefin, the rate of enzyme-product binding (from E + P to EP) was proven to be comparable with that of enzyme-substrate binding (from E + S to ES) (Yang, H., *Biochemistry* 2012, 51: 3839-3847). Cocrystallization of NDM-1 and hydrolyzed meropenem also clearly indicates the binding of product and enzyme (King, D.T., *JACS* 2012, 134:11362-11365). Such binding, however, does not affect the reliability of intermediate binding revealed in our crystal structures.

First, the chance of product rebinding to the enzyme in a soaked crystal is so low to be ignored. At the beginning of crystal soaking, substrate-enzyme binding is saturated very fast in the presence of excess carbapenem substrate (a molar ratio

of substrate/enzyme = 5:1 was used in our experiments). After the lactam-bond cleavage and the generation of an anionic intermediate, the reaction proceeding in the crystal may suspend at a stage of EI₁, EI₂ or EP, dependent on the chemical environment in a given crystal and the soaking duration. Only for the crystal with completed reaction and partial release of hydrolyzed product from the EP complex (i.e. from EP to E + P), the released product in solution may have chance to rebind the enzyme in the crystal (from E + P to EP). Owing to the much lower concentration of released product than that of non-hydrolyzed substrate in the solution, the chance of product rebinding is extremely low. Second, even if some product molecules indeed rebind to the enzyme in the crystal, the structure of newly formed EPs would be same as that of those with unreleased product still bound to the active site, due to the homogenous chemical environment in the crystal. In another word, product rebinding cannot give rise to a structure distinguishable from that generated from in situ hydrolysis.

I don't understand the reply to the NMR point – the NMR and crystallography measure different things the NMR measures the product not intermediates. Moreover at least some NMR on carbapenem hydrolysis has been published – see e.g. Smith et al. J. Pharmaceutical Sciences, 1990, 79, 732. (I do appreciate more modern NMR methods are used in the current paper.) Note there are many unassigned resonances in the NMR.

Reply: We already gave a detailed reply in the last response letter, which has clearly explained the necessity and significance of the NMR experiment in this study. In brief, NMR is a method that can provide structural information as a temporal average from the compound population in solution, but cannot provide time-resolved details of structural and chemical change during a reaction. Due to the weak signal/noise ratio and long collection time (2D spectrum in particular), NMR is an applicable technique for measuring the reaction product, but not the

intermediates that would decay within a time regime much shorter by several orders of magnitude than that suitable for NMR. The contribution of NMR spectroscopy in this study was to confirm that a β -diastereomeric product was exclusively generated from carbapenem hydrolysis as our crystallographic data suggested.

The article cited by the reviewer reports alkaline degradation of imipenem in solution, and is irrelevant with lactamase-catalyzed hydrolysis. As for the unassigned resonances in our NMR spectra, please refer to the above reply to this question.

The main issue is that without the time resolved aspect these are 'just' further structures of an already well-characterised enzyme class; note hydrolysed carbapenem structures have been reported with MBLs, see Wachino et al. Antimicrob. Agents Chemotherap. 2016, 60, 4274.

Reply: We disagree with this comment. Even without time-resolved data, our data do provide significant mechanistic insights into MBL-catalyzed carbapenem hydrolysis. The hydrolyzing mechanism of carbapenems has been proposed in a number of publications on the basis of spectroscopic data and QM/MM calculations, for examples Tioni, M. et al., JACS 2008, 130: 15852-15863, and Tripathi and Nair, ACS Catalysis 2015, 5: 2577-2586. Direct evidences for the proposed mechanism have been unfortunately lacking for quite a long time due to difficulties in structural characterization of the intermediates. Our study revealed three reaction species representing EI₁, EI₂ and EP with the missing bridging water in the crystal structures, and an exclusive β -diastereomeric product detected by NMR. These data provided direct evidences supporting the proposed reaction pathway, and further suggest a diverged mechanism in the step of intermediate protonation. We believe that many MBL researchers would be interested in the experimental findings reported in this manuscript, which

have been acknowledged by the other reviewers.

The article cited by the reviewer reported several crystal structures of a B3 subclass MBL in complex with hydrolyzed carbapenems and proposed a branched hydrolyzing mechanism different from B1 subclass MBLs, such as NDM-1. The experimental data described in that article, however, seem to have provided less mechanistic insight than ours as only complexes representing EP were revealed and no solution work was done, for example determination of the stereoselectivity between the released products by NMR measurement.

Overall, I'm sorry not to be more positive – in my view the structures (which are of interest to specialists especially with respect to the differences to the analogous complexes formed from penicillin and cephalosporin binding) could be reported in a specialist journal with appropriate caveats on what they represent – alternatively the authors could work to provide a more complete and convincing mechanistic picture including by doing time resolved crystallographic and solution work.

Reply: To our best knowledge, no time-resolved study for MBL-catalyzed carbapenem hydrolysis has been reported so far. This probably indicates that this system may not be a good research target for the current time-resolved technique, which is still in its infancy (Trincao J. et al., Dynamic structural science: recent developments in time-resolved spectroscopy and X-ray crystallography, *BIOCHEM SOC T* 2013, 41: 1260-1264.). In such a context, we do not think that time-resolve data are absolutely required for publication of this work in *Nature Communications*, since our study revealed precise structures of transient reaction intermediates and crucial reaction details such as the absence of the bridging water and an exclusive β -diastereomeric product. We hence believe that the experimental data reported in our manuscript provide significant mechanistic insights into carbapenem hydrolysis and are therefore of wide

interest for MBL researchers.

Responses to Reviewer #2 and #3's Comments

We appreciate for the positive comments and recommendation of manuscript acceptance from both reviewers, which are the best reward for our research. We also thank for the grammatical errors pointed out by Reviewer #3. All these errors have been corrected in the updated manuscript. In addition, the English of our manuscript was improved with the help from a colleague of us who is a native speaker.

Finally, we would like to thank again for the reviewing work done by all referees.